# Rapid transporter regulation prevents substrate flow traffic jams in boron transport

Naoyuki Sotta[1], Susan Duncan[2†], Mayuki Tanaka[1], Takafumi Sato[1], Athanasius FM Marée[2*], Toru Fujiwara[1*], Verônica A Grieneisen[2*]

[1]Graduate School of Agricultural and Life Sciences, The University of Tokyo, Tokyo, Japan; [2]Department of Computational and Systems Biology, John Innes Centre, Norwich, United Kingdom

**Abstract** Nutrient uptake by roots often involves substrate-dependent regulated nutrient transporters. For robust uptake, the system requires a regulatory circuit within cells and a collective, coordinated behaviour across the tissue. A paradigm for such systems is boron uptake, known for its directional transport and homeostasis, as boron is essential for plant growth but toxic at high concentrations. In *Arabidopsis thaliana*, boron uptake occurs via diffusion facilitators (NIPs) and exporters (BORs), each presenting distinct polarity. Intriguingly, although boron soil concentrations are homogenous and stable, both transporters manifest strikingly swift boron-dependent regulation. Through mathematical modelling, we demonstrate that slower regulation of these transporters leads to physiologically detrimental oscillatory behaviour. Cells become periodically exposed to potentially cytotoxic boron levels, and nutrient throughput to the xylem becomes hampered. We conclude that, while maintaining homeostasis, swift transporter regulation within a polarised tissue context is critical to prevent intrinsic traffic-jam like behaviour of nutrient flow.

DOI: https://doi.org/10.7554/eLife.27038.001

**\*For correspondence:**
Stan.Maree@jic.ac.uk (AFMM);
atorufu@mail.ecc.u-tokyo.ac.jp
(TF);
veronica.grieneisen@jic.ac.uk
(VAG)

**Present address:** [†]Earlham
Institute, Norwich, United
Kingdom

**Competing interests:** The
authors declare that no
competing interests exist.

**Reviewing editor:** Maria J
Harrison, Boyce Thompson
Institute for Plant Research,
United States

## Introduction

Robust growth and functioning of all living organisms, including plants, requires a well-balanced uptake of nutrients from the environment. However, nutrients are not always readily available in the environment at the organisms' optimal levels. This issue is exacerbated for sessile plants which take up essential minerals, required for their growth and functioning, from the soil in which the nutrient concentrations are mainly determined by elemental contents of igneous rocks. Most soils are either deficient or in excess of essential elements (*Roy et al., 2006*). Plants have therefore evolved to take up these essential minerals from the soil in a regulated manner over a wide range of concentrations through their roots, conveying the nutrients to the rest of the plant body via the vascular network. For a well-balanced uptake of nutrients from soils by roots, nutrient transport processes need to be regulated in a nutrient-dependent manner, which constitutes one of the essential processes for plant growth and crop production.

For roots to transport soil-derived nutrients into its central vascular tissues, which will ultimately transport it to other regions of the plant, a flow of nutrients has to be established which crosses several cell files, from the cells in contact with the medium (typically epidermal or lateral root cap cells) to the xylem. Boron is one of such essential nutrients for plants, critical for cell wall composition (*O'Neill et al., 2004*). Borate is cross-linking a pectic polysaccharide, Rhamnogalacturonan II, which makes it indispensable for tissue growth (*Kobayashi et al., 1996*). Only available in the soil, boron is transported to the xylem through an intricate system of polarly localised membrane transporters

**eLife digest** Every multicellular organism, including all plants and animals, faces the challenge of taking up the nutrients it needs and distributing them throughout its body. Plants absorb many nutrients including nitrogen and boron from the soil into their roots, often using tightly controlled processes that require energy to work. Plant roots contain several distinct layers of cells and the nutrients need to cross these layers to reach a channel at the centre of the root known as the xylem, which transports the nutrients to other parts of the plant.

Plants need boron to grow. However, high levels of this nutrient are toxic so plants have evolved to change the rate at which they absorb boron to optimize growth in different environments. When there is little boron in the soil, certain transporter proteins move to the surface of root cells to bring boron into the root more effectively. On the other hand, when plants grow in soils with high boron, their root cells have fewer of these transporters on their surfaces to prevent too much boron entering the plant.

This regulation of boron uptake appears logical, except for one detail: at any given location, the amount of boron in the soil is relatively stable and changes only very slowly. Why do plants invest energy in responding rapidly to the supply of a nutrient that changes so slowly in nature?

Sotta et al. used mathematics and experimental approaches to study boron uptake in a plant known as *Arabidopsis*. The work reveals that the plants ability to rapidly alter how efficiently boron moves into root cells actually serves to avoid internal "traffic jams" in boron transport. If the numbers of transporter proteins on the surface of root cells changed more slowly, individual cells would occasionally experience high levels of boron that would interfere with the movement of boron further into the root, causing a jam. Furthermore, these 'peaks' of boron could damage the individual cells they affect.

The findings of Sotta et al. reveal that, by being able to rapidly change the numbers of certain transporter proteins on the surface of root cells, plants can ensure they receive a steady supply of boron. This work suggests that to develop artificial systems that can adapt to changing surroundings, researchers will need to engineer solutions like those found in plants in order to avoid similar traffic jams in the systems. Along with considering how plants interact with their environment, studying how they avoid internal traffic jams in nutrient uptake may help efforts to alter plants, including crops, so that they grow better in harsh environments.

DOI: https://doi.org/10.7554/eLife.27038.002

that bring boron from the apoplast into the cytosol, and then back again into the apoplast (*Miwa and Fujiwara, 2010*). The boron transport mainly takes place in the form of boric acid and borate. At neutral pH, boron is mostly present in the form of boric acid (*Woods, 1996*).

Two families of transporters, BORs and NIPs, are involved in boron transport through the root. BOR proteins export boron from within the cell to the cell wall, while NIP proteins enhance bidirectional permeability. Both BORs and NIPs present polar localisation, often complementary. In the root, BOR1 (At2g47160) and BOR2 (At3g62270) locate on the inwards-facing membranes of the cell (*Takano et al., 2010*; *Miwa et al., 2013*), while NIP5;1 (At4g10380) locates on the outwards-facing membranes (*Takano et al., 2010*). Being polarised in such a manner, their combined action and localisation allows for a highly efficient boron uptake and transport to the xylem, even when boron is only available at very low concentrations in the medium. This transport system ensures that within the root tissue much higher boron concentrations can be reached than what is available in the medium, allowing for significant xylem loading which then provides sufficient boron for the growing shoot. Indeed, both *nip5;1* and *bor1, bor2* mutants present severe growth defects under low boron concentrations (*Noguchi et al., 1997*; *Takano et al., 2006-06*; *Miwa et al., 2013*). Such a coordinated and directed polar transport is reminiscent of the intricate polar auxin transport networks within the plant, although their axes of polarity are often perpendicular to one another (*Robert and Friml, 2009*).

Likewise, the boron transport system faces constraints: While boron is required for cell wall strength and stability, too high intracellular boron concentrations elicit DNA damage, growth retardation and eventually cell death (*Sakamoto et al., 2011-09*). Unsurprisingly, therefore, is has been

found that the protein levels of the boron transporters are regulated by boron itself, with both NIP5;1 and BOR1, 2 protein levels dropping at higher boron concentrations (*Takano et al., 2005*; *Takano et al., 2006-06*; *Takano et al., 2010*; *Miwa et al., 2013*). Such boron-dependent regulation is considered essential to allow for boron homoeostasis, preventing boron toxicity when boron availability is high, but allowing for efficient uptake when availability is low. Protein down-regulation takes place through two distinct mechanisms. In the case of NIP5;1 boron reduces protein levels via mRNA degradation (*Tanaka et al., 2011*; *Tanaka et al., 2016*), while in the case of BOR1, 2 the mechanism involves increased protein degradation (*Takano et al., 2005*; *Miwa et al., 2013*). Surprisingly, however, in both cases the down-regulation of boron transporters by boron occurs on a short time scale: when a plant is transferred from low to high levels of boron, swift downregulation of BORs (via protein degradation), and NIPs (via transcript degradation) is observed (*Takano et al., 2005*; *Tanaka et al., 2011*; *Tanaka et al., 2016*). The BOR1 degradation occurs through endocytosis, apparent 30 min after the transfer from low to high boron media. After two hours BOR1 has already mostly disappeared, suggesting that the half life of BOR1 is well below one hour (*Takano et al., 2005*). The half life of *NIP5;1* mRNA after the transfer from low to high boron media is 10–15 min (*Tanaka et al., 2011*). Such rapid time scales seem at odds with the expected natural variations of boron a plant would experience, as there is no evidence supporting considerable fluctuations in soil boron concentrations, neither spatially nor temporally. This is due to boron being available to plants as boric acid, which is highly water-soluble (solubility: 0.92 mol/L at 25°C). Consequently, boron is very mobile in the soil (*Nable et al., 1997*), rendering patchy heterogeneous boron levels throughout the soil neither stable nor probable. The main process that presumably would allow a plant growing at a fixed location to experience rising boron levels is through drought, a phenomenon which fails to account for the necessity of transporter down-regulation occurring on the order of minutes. Also watering of plants is not expected to quickly change the boron levels. *Keren and Bingham, 1985*, analysing the effects of solution-to-soil ratio on the boron concentration in soil water, propose that soil adsorption plays a role in buffering the fluctuations in boron concentration in soil water as a consequence of fluctuations in the water-soil ratio. Mathematical modelling was used to predict the boron concentrations in the soil solution from the water-to-soil ratio (*Keren, 1981*), revealing that boron concentration in the soil is robust against fluctuations in the water-to-soil ratio if soil adsorption is considered. From these in-depth studies, the picture consolidates that rapid fluctuations in boron concentration are indeed unlikely, or rare events even if they could occur under very specific and unlikely conditions. Given that there is no apparent necessity for swift regulation, it is thus surprising to find cost-ineffective down-regulation mechanisms through degradation of mRNA and protein underpinning this system, instead of more cost-effective down-regulation processes via transcriptional repression.

In short, down-regulation of boron transporters can be readily understood as a natural adaptive mechanism for plants to optimise growth and function at different geographical locations with varying natural boron concentrations. Nevertheless, it remains intriguing as to why plants have evolved such a swiftness in the regulation of these transporters. Puzzled by this behaviour, we questioned if other dynamical constraints are operating on polarised root tissue that could explain the need for these rapid timescales. To address this, we explored possible implications of dynamical transporter regulation in a parsimonious model that captures the nutrient flux across a small cross-section of a generic root tissue, with an explicit focus on the swiftness of the temporal regulation.

## A simple model for boron uptake to probe implications of boron transporter regulation swiftness

In *Arabidopsis thaliana*, boron is shuttled from the soil across the epidermis, cortex and endodermis into the stele, to finally be taken up into the xylem, mediated by the action of secondary active boron exporters (BORs) and the enhancement of permeability through NIPs. Previously, we analysed boron patterning by using a two-dimensional cross-sectional model of the entire Arabidopsis root meristem, considering all spatial nuances of transporter localisation, polarity and intensity, whilst neglecting transporter dynamics and regulation (*Shimotohno et al., 2015-04*). Instead, transporter levels and activity were static, fixed to the observed steady state transporter expression under constant boron medium condition of 0.3 μM. While neglecting regulation of transporter expression or activity, a characteristic boron pattern emerged on the level of the root, presenting highest concentrations at the stem cell niche. This boron profile gradually decreased longitudinally up to the start

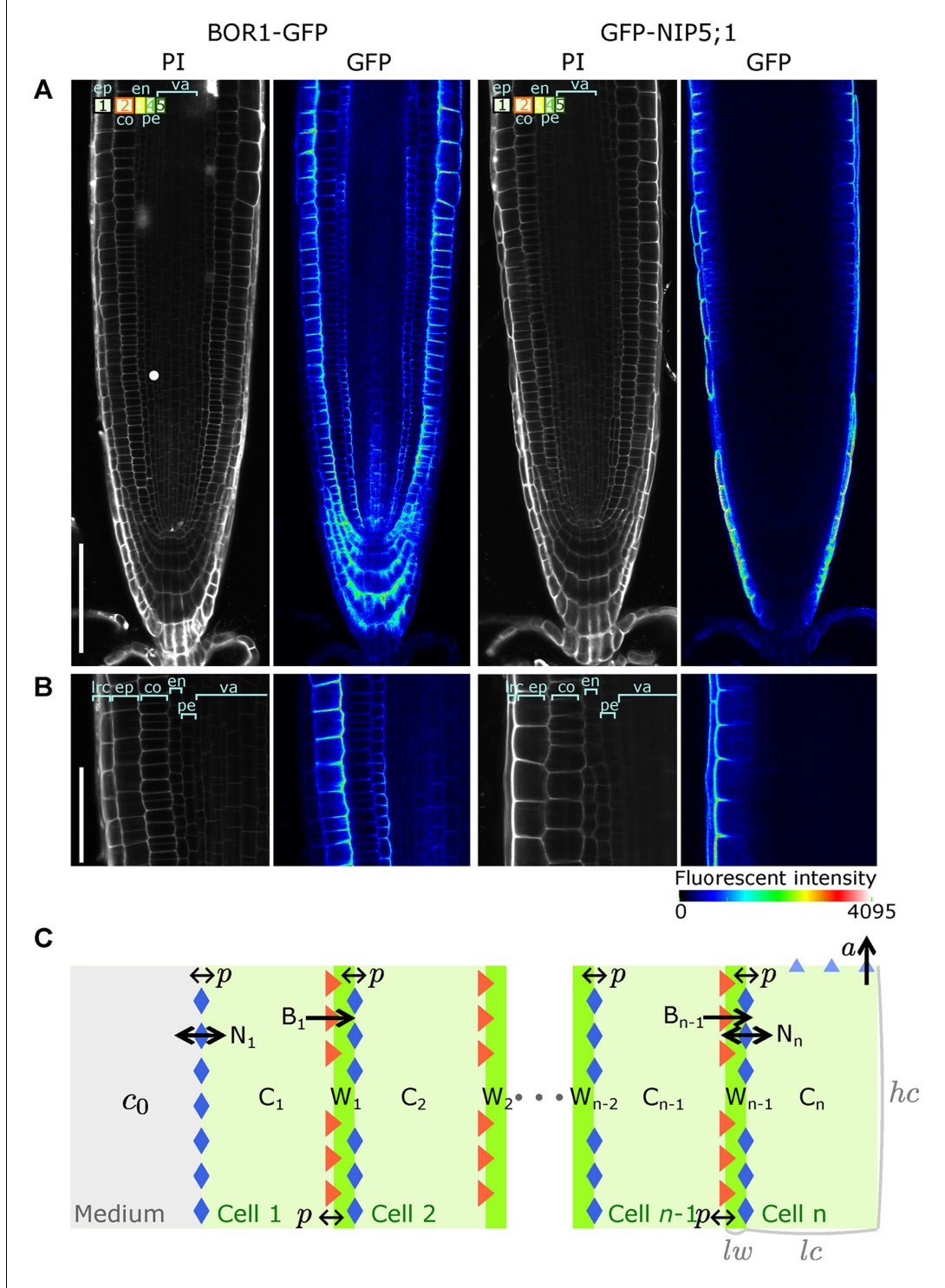

**Figure 1.** A polarised tissue model for nutrient boron transport across roots. (**A**) Confocal microscope images of BOR1-GFP and GFP-NIP5;1 localisation in *A. thaliana* roots, revealing polar localisation of both transporters, facing inwards and outwards, respectively. Cell file identities are shown in the top-left with numbers representing, in increasing order, cell files from the outermost to the innermost cells, such as ep, epidermis; co, cortex; en, endodermis; va, vasculature. (**B**) Detailed view along a transversal section of the tissue at the proximal meristem. lrc, lateral root cap. (**C**) Schematic diagram of the NIP-BOR boron transport model, consisting of a transversal root cross-section composed of $n$ cells between the medium and the xylem. For a simplified *A. thaliana* model, we consider $n = 4$. Depicted is a generic root model consisting of $n$ cell files, not showing all cells and cell walls in between $W_2$ and $W_{n-2}$ for illustration purposes. The model includes intracellular and apoplastic compartments, as well as membrane-based properties

*Figure 1 continued on next page*

*Figure 1 continued*

such as transporter activity and background permeability rates. *lw*: cell wall width; *lc*: cell width; *hc*: cell height. For a full description of the parameters, see **Table 1**.

DOI: https://doi.org/10.7554/eLife.27038.003

The following figure supplements are available for figure 1:

**Figure supplement 1.** Promoter activity of *NIP5;1* in roots.

DOI: https://doi.org/10.7554/eLife.27038.004

**Figure supplement 2.** Different boron sensitivity of NIP5;1 and BOR1 with respect to the boron concentration.

DOI: https://doi.org/10.7554/eLife.27038.005

of the differentiated tissues. The computationally simulated characteristic boron distribution, which we confirmed experimentally through LA-ICP-MS, strongly indicates that the mature root tissue is involved mainly in transporting boron from the soil into the xylem, while the distal meristematic tissues have a differential boron transport function, namely, to locally provide higher boron levels for fuelling local growth (i.e., cell wall material) at those regions. In that study, however, we did not address the dynamics of the transporter regulation nor how nutrient homeostasis is achieved. Here, we investigate the dynamics of the transporters and their mutual feedback with the generated boron distribution. We do so by spatially focussing on the differentiated tissue region involved in xylem-uptake.

At the elongation and differentiation zone of the root, transporters possess a striking polar expression; BORs are located at the inside-facing membranes of the cells, while NIPs are concentrated to the outside facets (see **Figure 1A,B**). Although protein levels can present strong variation between cell files, the transcription of the transporters takes place throughout the entire root tissue, even for NIP5;1 (**Figure 1—figure supplement 1**). We therefore use the simplifying assumption that all cell files are intrinsically the same in respect to the potential of expressing the transporters, and variations arise solely as a consequence of the nutrient distributions.

Our analysis focuses on the transversal nutrient flow through the root that results from boron entering and leaving the different cell files transversally. Effectively, in our simple model, we only consider a single row of cells and, for simplicity, only four cell files over which boron is transported from the soil into the xylem (**Figure 1C**). To capture the dynamics of boron transport and transporter activities in root tissue, the model's variables are the boron concentration in cells (C) and cell walls (W), and transporter activities of NIPs (N) and BORs (B) for each individual cell (n). The mutual dependency between these variables is described using ordinary differential equations (ODEs). For each cell file of the model, the transporters are produced at an equal rate, have the same transport potential, and are regulated in the same way. The modelled cells loosely map to the outermost epidermis (cell 1), the cortex (cell 2), the endodermis (cell 3), and finally to the pericycle/vasculature (cell 4). The last cell of the row is endowed with an upward convective flow, which captures the shootwards convective flow of the xylem. For reasons of simplicity, we consider these four cells to have equal dimensions, and make the same assumption for all intermediate cell walls (**Figure 1C**).

Given that transporter regulation can be observed in response to varying the boron conditions in the medium (**Takano et al., 2005**, Takano et al., 2006-06**Takano et al., 2006-06**, **2010**; **Miwa et al., 2013**), (**Figure 1—figure supplement 2**), boron sensing underpinning those responses could either involve measurements of the intracellular concentration or involve measurements of the intercellular concentration (i.e., the levels within the apoplast). In the case of NIP5;1, we recently revealed that *NIP5;1* mRNA degradation is triggered by boron-dependent ribosome stalling during the translation process (**Tanaka et al., 2016**). The boron-dependent degradation was also observed in an in vitro system without cell wall fraction, which suggests that the sensing mechanism depends on the cytoplasmic boron concentration. For the BORs the precise subcellular location of the boron sensing has not yet been determined, but it is reasonable to assume that the regulation is also responding to cytosolic boron levels, given that the purpose of the nutrient homeostasis is to keep the nutrient levels within the cell within bounds, and to this end directly sensing the cytoplasmic boron concentration is beneficial. In short, given the current knowledge, cytosolic boron sensing is the most parsimonious assumption.

Hence, our model assumes that all the boron transporters of a cell are regulated by the cytosolic boron concentration within that respective cell. Note that we loosely use the term 'boron' in our

model to refer to both boric acid as well as borate. They can be found in a dynamic chemical equilibrium, $B(OH)_3 + H_2O \Longleftrightarrow B(OH)_4^- + H^+$, with a $pK$ value of 9.24 (**Woods, 1996**). Given that this process does not involve any enzyme kinetics, this distinction is not explicitly considered in our model. Moreover, we solely focus on the import of soluble boron (both boric acid and borate), not considering any boron that is bound to the cell wall.

The temporal dynamics of the soluble boron concentration within the cytosol ($C_i$) and in the cell walls ($W_i$, being the cell wall adjacent to the inward facing facet of cell $i$) is determined by their inflow and outflow from and to neighbouring compartments:

$$\dot{C}_i = \{(p + N_i)(W_{i-1} - C_i) - B_i C_i + p(W_i - C_i)\}\frac{1}{lc}, \tag{1}$$

$$\dot{W}_i = \{p(C_i - W_i) + (p + N_{i+1})(C_{i+1} - W_i) + B_i C_i\}\frac{1}{lw}. \tag{2}$$

Here, $N_i$ and $B_i$ represent boron permeabilities due to, respectively, the bi-directional transport by NIPs and the directional transport by BORs within cell $i$. Transporter-independent boron permeability through the plasma-membrane is also taken into account, described by parameter $p$. To capture the tissue context, boundary conditions are set at both extremities of the transversal cell series. The outer cell wall of the first cell (cell 1) is in contact with medium. Given the rapid diffusion rate of boric acid in water, we assume that the boron concentration in the outermost cell wall ($W_0$) is the same as in the medium, constant over time. The xylem transport that occurs from roots to upper tissue is represented by a removal term, with rate $a$, attributed to the innermost cell ($i = n$) of the cell row:

$$\dot{C}_n = (p + N_n)(W_{n-1} - C_n)\frac{1}{lc} - aC_n\frac{1}{hc}. \tag{3}$$

Assuming that transport activities are proportional to protein concentration, it follows that the dynamical regulation of NIP and BOR permeability ($N_i$ and $B_i$) are in direct proportion to the dynamics of their proteins (we will later show that our main insights do not depend on the assumption that the BOR transporter does not saturate under the range of concentrations here treated). It was therefore not necessary to introduce explicit variables for protein concentrations. Instead we directly use permeability, representing the protein concentrations and their regulation. The protein concentration, and thus the effective permeability, is determined by production and degradation rates.

As mentioned, accumulation of *NIP5;1* mRNA is regulated through boron-dependent mRNA degradation. This suggest that the production of NIP5;1 protein is boron dependent. On the other hand, degradation of NIP5;1 protein is not found to be boron dependent (**Takano et al., 2010**; **Tanaka et al., 2016**). Thus, the levels of $N_i$ vary due to a production term that is dependent on cytosolic boron concentration and a constant degradation term:

$$\dot{N}_i = \alpha_N \frac{k_N^{n_N}}{k_N^{n_N} + C_i^{n_N}} - \xi_N N_i, \tag{4}$$

where $\alpha_N$ is the production rate, $\xi_N$ is degradation rate, and $n_N$ is the Hill coefficient capturing the boron-dependent inhibitory regulation.

Accordingly, given that BOR protein levels are regulated through boron-dependent degradation (**Takano et al., 2005**), but there is no evidence of boron dependent production, $B_i$ varies over time due to constant production and boron-dependent degradation terms:

$$\dot{B}_i = \alpha_B - \xi_B \left(1 + d_B \frac{C_i^{n_B}}{k_B^{n_B} + C_i^{n_B}}\right) B_i, \tag{5}$$

where $\alpha_B$ is the production rate, $\xi_B$ the basal degradation rate, $d_B$ the maximum boron-independent degradation rate, and $n_B$ the Hill coefficient capturing the boron-dependent inhibitory regulation.

We wish to use this model to understand the significance, if any, of the swift transporter regulation. To assess the effects of regulation swiftness on transport, we therefore introduce a time-scaling factor $\epsilon$, which multiplies both the entire $\dot{N}_i$ and $\dot{B}_i$ terms (Equations 6 and 7). To align ourselves with the current experimental evidence that indicates swift regulation of NIP5;1 on the mRNA level and

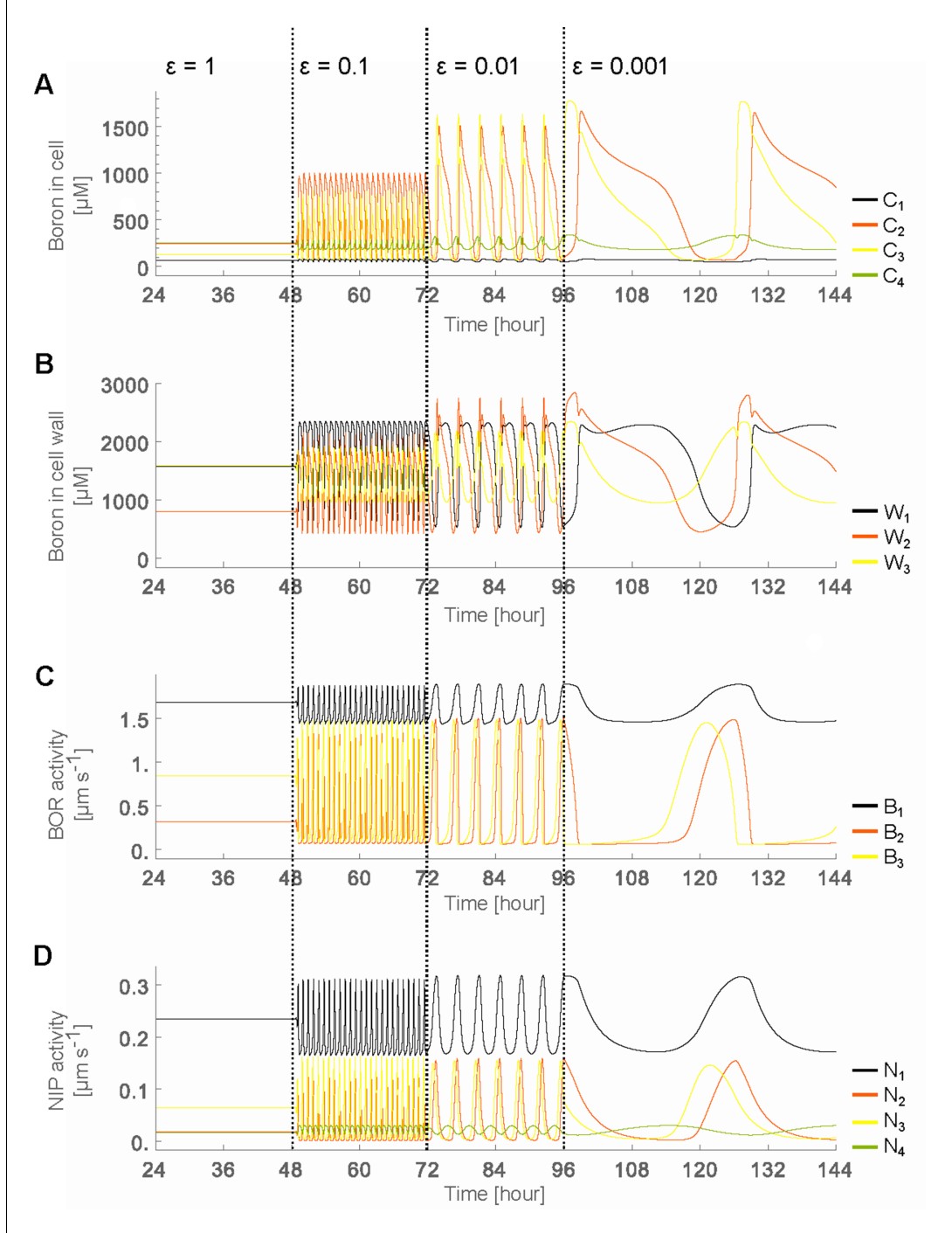

**Figure 2.** Oscillatory behaviour arises due to decreased transporter regulation swiftness. Time development in the four-cell NIP-BOR model of the boron concentration in the cells (**A**) and cell walls (**B**), and of the transporter activity of BOR (**C**) and NIP (**D**). The simulations are started using the default parameter setting (**Table 1**). At the time points 48, 72 and 96 hr, the parameters determining the transporter regulation dynamics ($\epsilon_N$ and $\epsilon_B$) were each time reduced to one tenth of their previous value. $C_1$ to $C_4$ are the boron concentrations in the outermost cell 1 up to the innnermost cell 4. $W_i$ represents boron concentration in cell wall fraction between cell $i$ and cell $i+1$ and $N_i$ are transporter activity of BOR and NIP in cell $i$, respectively.
DOI: https://doi.org/10.7554/eLife.27038.006

The following figure supplement is available for figure 2:

**Figure supplement 1.** Robustness of behaviour regarding choice of $k_B$ parameter.
DOI: https://doi.org/10.7554/eLife.27038.007

of BORs on the protein level, we modulate the production rate and degradation rate for NIP5;1 and BOR1, respectively. However, changing production rates of NIPs through a factor $\epsilon$ without changing the degradation rate would result in changes of the overall protein concentration which might influence the system's behaviour, not due to the swiftness of the regulation but due to the changes in equilibrium protein concentration. This also holds for the BOR regulation. Thus, to avoid such possible undesired effects, the time-scaling factor multiplies the entire right-hand side of the dynamics:

$$\dot{N}_i = \epsilon_N \left( \alpha_N \frac{k_N^{n_N}}{k_N^{n_N} + C_i^{n_N}} - \xi_N N_i \right), \tag{6}$$

$$\dot{B}_i = \epsilon_B \left\{ \alpha_B - \xi_B \left( 1 + d_B \frac{C_i^{n_B}}{k_B^{n_B} + C_i^{n_B}} \right) B_i \right\}. \tag{7}$$

We define $\epsilon = 1$ as representing the swiftness of regulation in the wild type scenario, while reducing $\epsilon$ allows us to simulate behaviours that would occur with less rapid regulatory dynamics, and increasing $\epsilon$ allows us to consider even higher swiftness of regulation than experimentally observed. Note that the regulation of the transporter activity by cytosolic boron concentrations occurs through a sigmoidal relationship, in which $k_B$ and $k_N$ are the boron concentrations at the half maximum of the Hill function, defining the sensitivity of the response to the boron concentration. We set $k_B \simeq 30 k_N$, consistent with our observations that the sensitivity of BOR expression is much less than that of NIP5;1 (*Figure 1–Figure supplement 2*), but will also show that our results do not require large differences between these two parameters.

We have assessed what is the biologically acceptable range for each parameter, based on the available literature (see Material and methods for details). Default parameter values were chosen to lie within these valid ranges, and are given in *Table 1*.

## From steady state flows to oscillatory dynamics

When regulation swiftness lies within the experimentally observed regime ($\epsilon_B = \epsilon_N = \epsilon = 1$), a constant flow of boron that is transported across the root ensues (*Figure 2*). Boron concentrations within the cells ($C_n$) remain constant over time. This steady state behaviour, however, is disrupted when the transporter regulation swiftness is decreased ten-fold ($\epsilon = 0.1$). Oscillations in the boron concentration arise in all cells, but most strikingly in the endodermis (cell 3) and cortex (cell 2) (*Figure 2A*), as well as in the cell walls (*Figure 2B*). Decreasing the transporters' regulation swiftness even further ($\epsilon = 0.01, 0.001$) consistently enhances the amplitude of such oscillations, and enlarges their periods (*Figure 2A,B*). We conclude that even when environmental boron levels ($C_0$) and xylem activity ($a$) remain constant, simply deviating from the rapid experimentally observed transporter's regulation swiftness is sufficient to push the boron transport system into an oscillatory dynamics.

The reason the cells undergo such dramatic concentration variations at lower regulation swiftness can be intuitively linked to the accompanying changes in transporter levels which also ensue (*Figure 2C,D*). They are initially triggered by minute fluctuations in the concentrations, but then cause increasingly larger changes in cytosolic and apoplastic concentrations. It is not immediately clear, however, how it depends on either the NIP or BOR transporter regulation, why regulation swiftness exacerbates oscillation amplitudes and how their interdependencies either produce stable flows or unstable oscillations that propagate throughout the tissue. What we can already conclude though, is that within a polarised tissue rapid transport dynamics are necessary to ensure flux homeostasis and avoid instabilities in concentration levels. But to fully understand the process we will first observe how in the model the steady flow regime behaves in terms of transporter expression levels when we assume that the regulatory dynamics are as swift as experimentally observed.

## Parsimonious model, under wild type dynamical settings, generates stable transporter expression

Given that our question is focused on the possible advantages of swift transporter regulation within a polarised tissue context, our model purposely ignored details of tissue-specific regulation, using the simplifying assumption that all cells are endowed with the same potential of expressing transporters. This assumption may initially seem at odds with the actual distinct GFP levels of the GFP-tagged proteins (*Figure 1B*), which qualitatively indicates stronger BOR1 expression in the epidermis

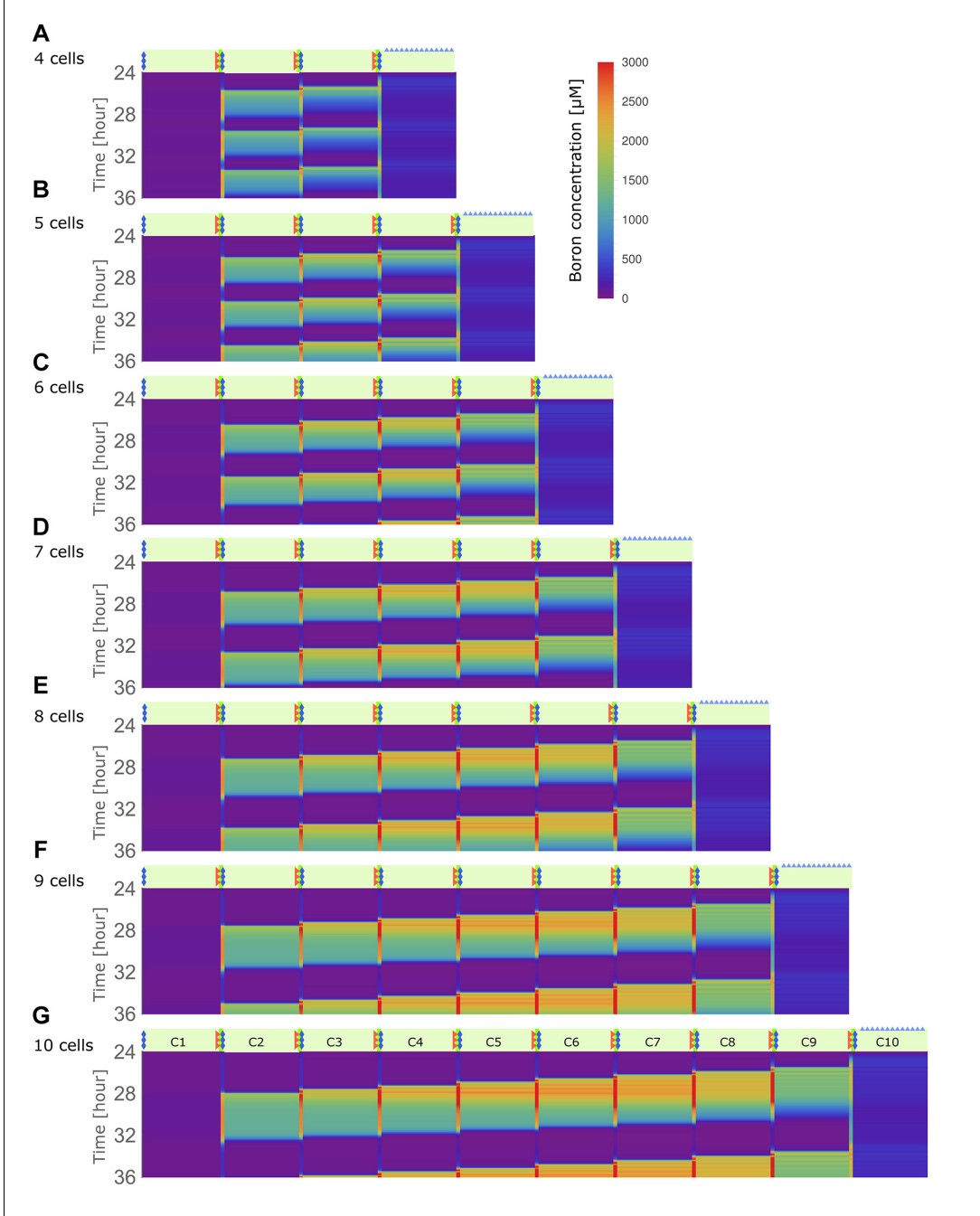

**Figure 3.** Traffic jam-like behaviour occurs for various tissue sizes. (A–G) Kymographs of boron concentration in NIP-BOR models consisting of four (**A**) up to ten (**G**) cells. Boron concentrations in cells and cell walls are represented through a heat map colouring with development over time displayed vertically. Boron concentrations exceeding 3000 μM are displayed using the same red colour. Regions of high concentration can be observed to displace to the left, contrary to the direction of boron flow due to the transporter activity. Default parameters were used (**Table 1**), with exception of the transporter regulation swiftness, $\epsilon_N$ and $\epsilon_B$, which were both set to 0.01. Above each kymograph a schematic diagram of the tissue structure is shown, corresponding to that of **Figure 1C**.

DOI: https://doi.org/10.7554/eLife.27038.008

The following figure supplement is available for figure 3:

**Figure supplement 1.** Kymographs of boron concentration in a 100-cell ring model.

DOI: https://doi.org/10.7554/eLife.27038.009

and endodermal files and pronounced NIP5;1 expression in the outermost cell file (lateral root cap or epidermis) only. It was therefore interesting to observe that our parsimonious model could already account for general features of the the observed transporters' expression patterns (*Figure 2C,D*) solely as a consequence of the relative positioning of the cells within the cell row: in the model, NIP only becomes highly expressed in the outermost cell, the epidermis ($N_1$, representing NIP5;1 in cell 1, is by far the highest variable in the steady flow case). These relative levels are in qualitative agreement with the experimental findings regarding NIP5;1. Moreover, in contrast to this strong expression in the epidermis, in the model NIPs are present at much lower amounts in all the other inner cell files (cell 2, cell 3 and cell 4), again qualitatively similar to the experimental observations. These model results can be understood as follows. Although all in silico cells share the same intrinsic properties, as a consequence of NIPs acting as permeability facilitators rather than directional transporters, it is impossible for the cell file in contact with the medium (cell 1, the outermost cell file, that is, epidermal cell) to reach higher concentrations than the level in the medium. Only the next cell inwards, cell 2, is capable of reaching higher levels, due to the directed polar action of the BORs. The inevitable low boron levels in the epidermis give rise to the observed lack of NIP downregulation within the first cell file, and hence to its preferential expression.

The expression patterns of the active exporters, BORs, are quite different from NIP5;1, but again there are similarities between the model results (B values) and confocal images, with BORs distinctly expressed in the epidermis (cell 1) at the highest levels, in the cortex (cell 2) at high levels, whilst in the endodermis it is relatively weaker (cell 3) (*Figure 1B*). The model's qualitative correspondence

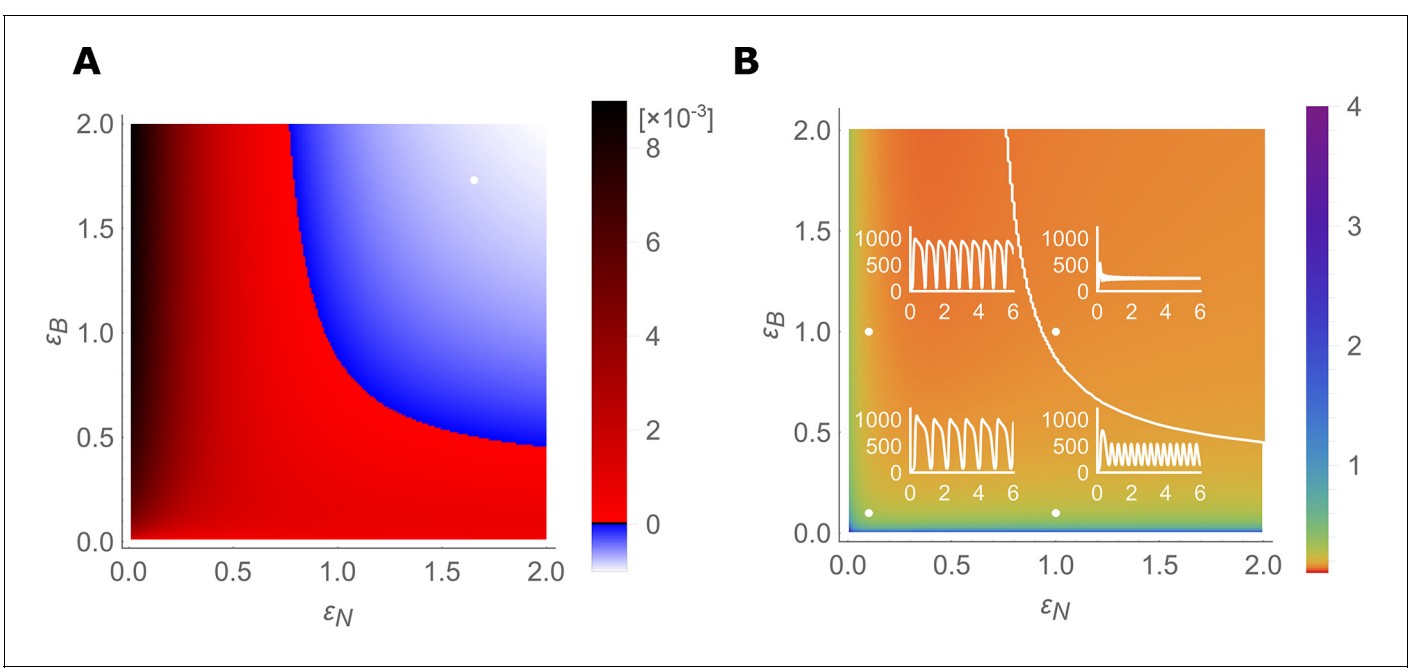

**Figure 4.** Stable- and unstable-flow regimes and their parameter dependencies. (**A**) Diagram showing combinations of transporter regulation swiftness ($\epsilon_N$–$\epsilon_B$ plane) for the four-cell NIP-BOR model for which stable flows (blue) or oscillations, that is, 'traffic jams', (red) ensue. (**B**) Oscillatory periods of the boron concentration variation. The white curved line represents the boundary between the stable and the unstable region, equivalent to the boundary between the blue and red area in (**A**). Four insets plot $C_2$ [µM] over time [hour] for the indicated parameter values (white dots), illustrating the behaviour of the model at those points within the $\epsilon_N$–$\epsilon_B$ plane.

DOI: https://doi.org/10.7554/eLife.27038.010

The following figure supplements are available for figure 4:

**Figure supplement 1.** Oscillation period dependency on transporter regulation swiftness.
DOI: https://doi.org/10.7554/eLife.27038.011

**Figure supplement 2.** Stable and unstable flow regimes and their parameter dependencies in the five-cell model.
DOI: https://doi.org/10.7554/eLife.27038.012

**Figure supplement 3.** Traffic-jam-like behaviour and saturation of the BOR1 transporter permeability.
DOI: https://doi.org/10.7554/eLife.27038.013

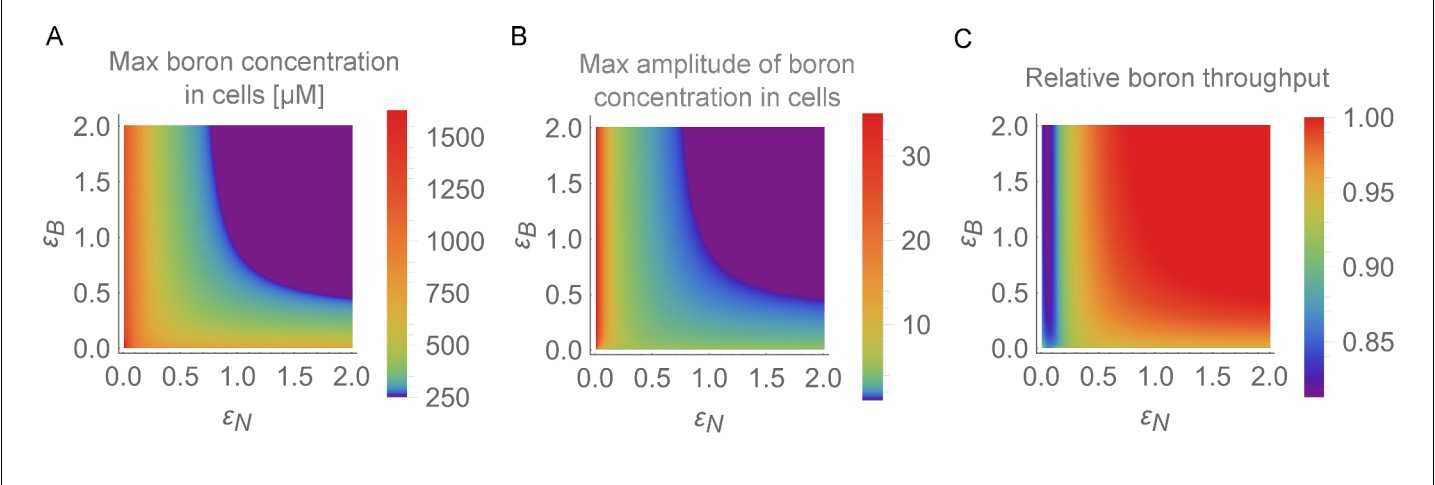

**Figure 5.** Physiological detrimental impact of the oscillations for the root. In the BOR-NIP model with 4 cells, physiological output for the simulation period between 24 and 48 hr was displayed on the $\epsilon_N$–$\epsilon_B$ plane. (**A**) Maximum boron concentration in any cell. (**B**) Maximum amplitude variation in boron concentration for any cell (highest boron concentration/lowest boron concentration). (**C**) Average throughput, measured as the output from the last cell, which represents the xylem, between 24 and 48 hr of simulation.
DOI: https://doi.org/10.7554/eLife.27038.014

between the transporter expression patterns and those experimentally observed suggests that concentration-dependent regulation is at least sufficient for such expression patterns to arise. Corroborating this, GUS expression under the *NIP5;1* promoter is indeed present in the inner root tissues, albeit the NIP5;1 protein levels in those cells are very low, supporting the notion that these files are likely responsive and capable of expressing NIP5;1, but inhibited to do so by the boron levels that arise due to the cells' positioning within the tissue (*Figure 1—figure supplement 1*). However, these results, while suggesting sufficiency of our parsimonious assumptions in relation to observed transporter levels, do not offer evidence that tissue-specific regulation is not occurring. At most, the qualitative matches justify, at first instance, that the model is kept simple, without the need of introducing ad hoc tissue-specific dependencies, when exploring the effects transporter regulation dynamics. Furthermore, these results indicate that under our experimentally observed parameter values, the system presents steady nutrient throughput and constant cellular concentrations and transporter levels over time.

## The spatial nature of the unstable flows: traffic-jam-like behaviour

We next investigated more carefully the spatial-temporal characteristics of the oscillatory behaviour that arises when transporter swiftness is decreased (*Figure 3A*). Kymographs make it visually clear that oscillations in the individual cells are in fact spatially coupled, manifested as a boron wave that propagates backwards over the tissue, that is, the pulse moves in the direction contrary to the nutrient flow itself. The initiation of the instability, as seen by the first peak in concentration, occurs close to the xylem (i.e., the last cell, here, cell 4), followed by a strong rise in endodermal concentrations (in cell 3) around 90 min later. Due to boron's inhibitory influence on the transporters, this intracellular rise leads to a decrease in both NIPs and BORs within that cell. This transporter downregulation causes a drop in boron throughput across that cell, leading to an accumulation of boron in the adjacent, outward facing, cell wall. Background permeability rates along the plasma membrane allow this apoplastic rise in concentrations to trigger an increase boron concentrations in the next outermost neighbouring cell, in this case, the cortex cell (cell 2). Cortex concentration levels thus rise, again triggering a shutdown of transporters, causing the same process to ensue in a spatially coupled manner in the outermost cell, the epidermis. In the epidermis, however, boron levels can never rise above the soil concentration levels, as discussed previously, such that the backward travelling wave loses its amplitude and terminates. The overall process is observed to be cyclical, with levels rising again close to the xylem, until a next wave is triggered in the innermost cells (*Video 2*).

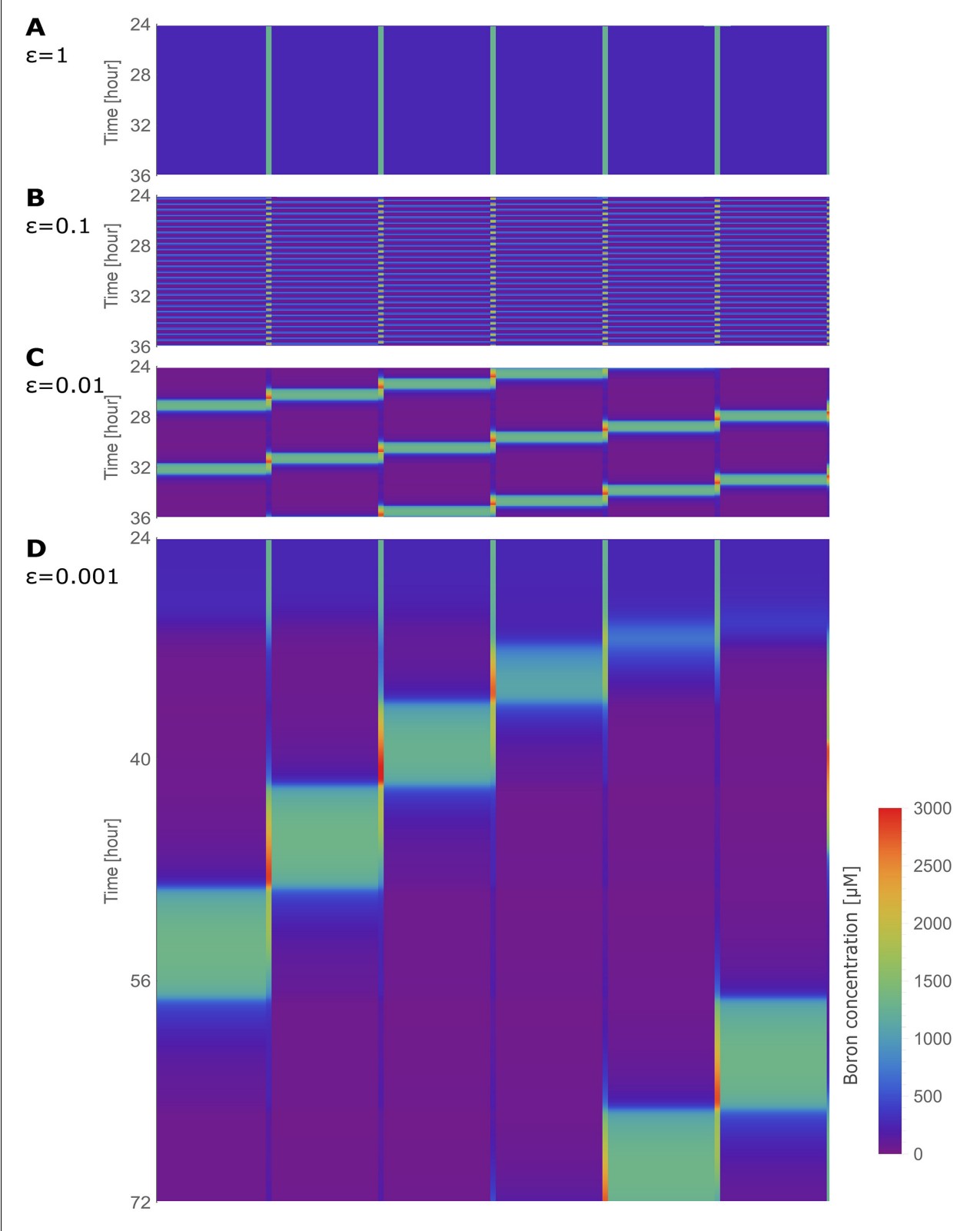

**Figure 6.** Traffic jam-like behaviour in the absence of bottlenecks: the tissue positioned in a ring. Kymographs of the boron concentration in the cells and cell walls of the six-cell NIP-BOR ring model, represented by heat map colours and time development shown progressing vertically. The default parameters were used (**Table 1**), with $\epsilon_N$ and $\epsilon_B$ varied as specified for each kymograph.
DOI: https://doi.org/10.7554/eLife.27038.015

Moreover, we found that such traffic-jam-like behaviour is robust when smaller differences between $k_B$ and $k_N$ are considered (*Figure 2—figure supplement 1A–C*), as long as $k_B$ is sufficiently large as not to too strongly interrupt boron throughput through the tissue altogether (*Figure 2—figure supplement 1D*). Obviously, if $k_B$ is too low, and traffic-jam-like behaviour are prevented, the plant would also not be able to take up boron, thus, this would be a parameter setting that is biologically irrelevant. Therefore, although our interpretation of the experimental results (*Figure 1—figure supplement 2*) suggest larger differences between $k_B$ and $k_N$, given that these are indirect estimates that rely on underlying assumptions, this robustness analysis shows that, within a biologically reasonable window, our results hold even when $k_B$ were to be smaller.

We next wondered if this traffic-jam-like effect dissipates over larger tissues. Our four-cell model is a simplified representation of *A. thaliana*, which has less cell files between the epidermis and the xylem than most other roots of experimental plant models. However, if one distinguishes the pericycle from the vasculature, a five-cell model would be more appropriate. We show that the variables of the five-cell model present similar dynamics as our default model when BOR transporter regulation is lowered to 400 µM (*Figure 4—figure supplement 2A–D*). Lowering $k_B$ slightly stabilises the dynamics (as shown in *Figure 2—figure supplement 1*), which counteracts increasing destabilisation of the constant flow regime when the number of cell files in the tissue increases. Conversely, given that this renders smaller tissues (such as our default four-cell model), more robust against traffic-jam-like behaviour, our default setting should be regarded as a worse-case scenario for such dynamics to occur, rather than being a special case. To further appreciate tissue-size dependencies, and explore the generality of these effects for other plants which might have highly deviating cell file numbers, we gradually increased the in silico tissue from 4 to 10 cell files. *Figure 3B–G* shows that the phenomenon is conserved irrespective of the number of cells between the medium and the convective xylem flux. Larger transversal tissue segments are able to generate higher wave amplitudes, but in all cases the peak in boron propagates contrary to the nutrient flow with approximately the same speed. The same phenomenon was still observed even when the model contained 100 cell files (*Figure 3—figure supplement 1*), suggesting that large tissues do not prevent these effects from occurring, but rather increase its likelihood.

In short, when transporter regulation swiftness is sufficiently slow, due to the downregulation of transporters under the control of increased boron concentrations, a traffic jam-like behaviour emerges: a high boron concentration peak appears, correlating with locally lower transport rates, triggering a low-transport high-boron wave that back-propagates from cell to cell in the direction contrary to the transport polarisation direction within the tissue. This occurs independently of the size of the plant tissue under consideration.

We next studied the importance of the relative dynamics of BOR and NIP regulation to trigger these phenomena, that is, which of the processes have to be sufficiently slow.

## Traffic jam-like behaviour depends on swift regulation of both transporters

Under what conditions and parameter values for transporter regulation rates does the traffic jam phenomenon manifest itself? In our previous simulations, we simultaneously decreased the swiftness of both the NIP and the BOR regulation. Oscillations in boron concentration then arose due to a change in stability of both the flow and steady state of boron concentrations. We next probed the system's stability while varying the regulation swiftness of the NIPs and BORs independently. This was done by analysing the equilibrium of the simplified, *A. thaliana* inspired, four-cell model. We linearised the ODEs around the equilibrium using a Taylor expansion, and then evaluated the stability of the equilibrium in terms of the largest eigenvalue, for 40,000 different parameter combinations. The phase-portrait that emerges (*Figure 4A*) shows a large region in which the system becomes unstable and oscillations arise (indicated in red), as well as a region in which sufficient regulation swiftness ensures that the system is stable and oscillations do not develop (indicated in blue). For the stable (blue) parameter settings, any perturbation dampens out (*Figure 4B*), whereas for parameter settings within the red region, oscillations dominate. The frequency of the oscillations is determined by a relative lack of swift regulation for both transporters, as shown by the colour map in *Figure 4*. This phase diagram is qualitatively unaffected for the five-cell model at $k_B =$ 400 µM (*Figure 4—figure supplement 2F*). Furthermore, we also checked if saturation in the activity of the BOR transporter might obstruct or stabilise the system against the appearance of the oscillations. (The

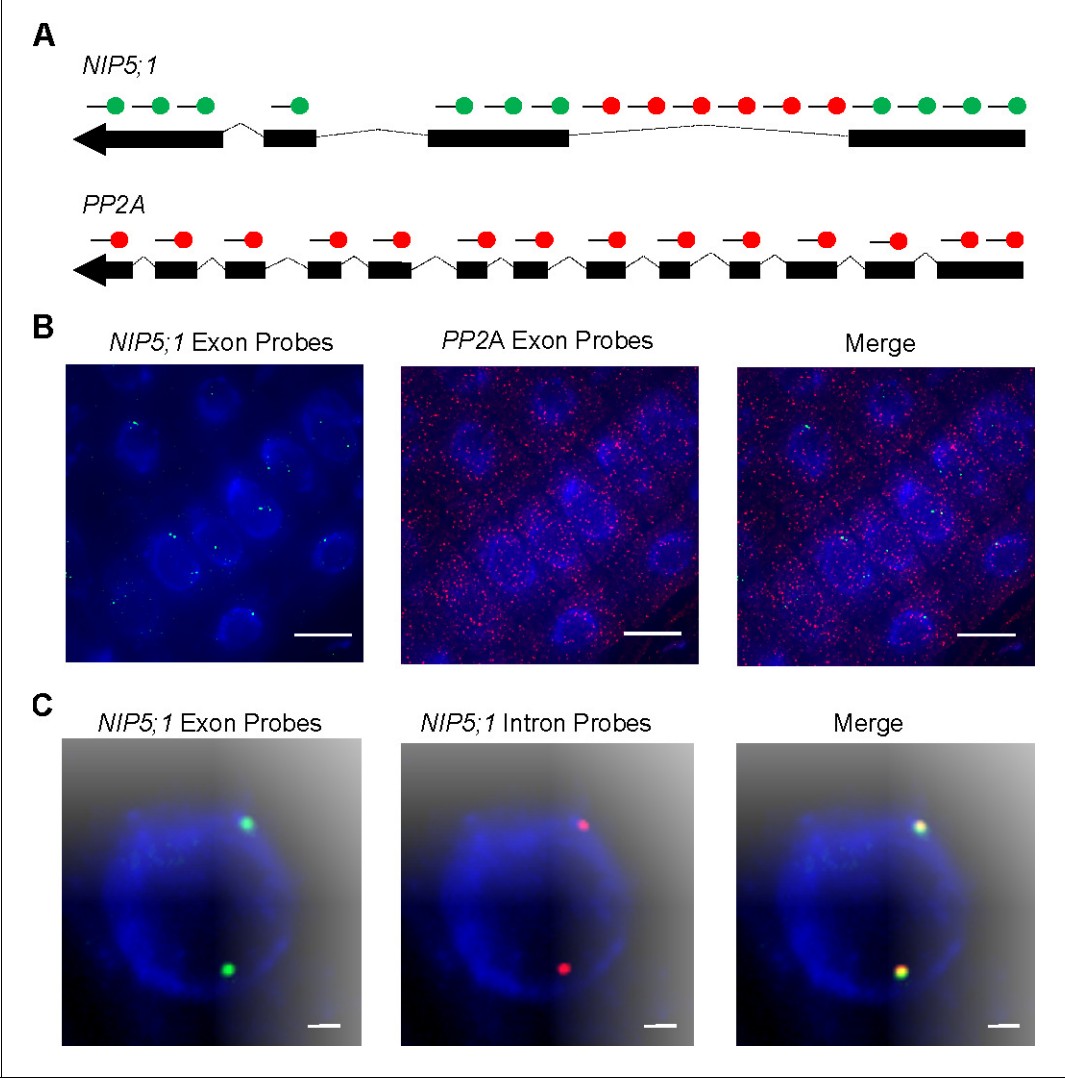

**Figure 7.** Nascent *NIP5;1* RNA accumulates at sites of transcription. (**A**) Schematic of probes used to detect *NIP5;1* intron 1 (red), *NIP5;1* exonic (green) and *PP2A* exonic (red) RNA. (**B**) Representative images of cells showing *NIP5;1* (green) and *PP2A* (red) exonic RNA distributions. (**C**) Representative images showing co localisation of exons (green) and intron 1 (red) regions from *NIP5;1* nascent transcripts. DNA labelled with DAPI (blue). Scale bar: 10 µm in (B) and 1 µm in (C).

DOI: https://doi.org/10.7554/eLife.27038.017

The following figure supplement is available for figure 7:

**Figure supplement 1.** Mature and pre-mRNA abundance of *NIP5;1* in root cells.

DOI: https://doi.org/10.7554/eLife.27038.018

NIPs, being a channel, should be less prone to undergo concentration-dependent saturation effects in permeability.) Contrary to this, we found that such a saturation in permeability of the BOR transporter does not change the likelihood or transporter-response dependency to present traffic-jam-like behaviour, but rather, when they arise due to slow dynamics, the oscillations present much higher amplitudes in concentration variations, with peak boron levels becoming more than twice as high due to the transport saturation (*Figure 4—figure supplement 3*).

From this combined analysis it becomes apparent that the parameter values we derived to capture wild type dynamics of transporter regulation (i.e., point 1,1) fall into the stable regime. Hence, the wild type system is robust against perturbations and does not present traffic jam-like behaviour. However, independent small reductions from these values in either NIP or BOR regulation rate can cause the tissue to experience oscillations. Not only does the analysis reveal that both transporters

are required to present a rapid response, and that this is independent of the BOR-dependent fluxes becoming saturated, but their combined swiftness is synergistic: if one transporter is extremely fast, the other can be bit slower than what is otherwise needed. However, for any parameter setting, a certain speed in responsiveness needs to present to prevent oscillations.

## Physiological implications of flow instabilities for the root system

Traffic jams are commonly experienced by vehicle drivers in urban areas. In terms of human transport, traffic jams are an undesirable outcome as they reduce the throughput through the highway and increase the time to reach the destination. Traffic jams displace backwards in space (contrary to the flow) while giving rise to increased car densities at the specific location of the traffic jam (*Kesting and Treiber, 2013*). All those properties can also observed in our plant tissue model, namely diminished throughput of boron from the soil to the xylem and backwards propagating pulses of locally increased boron concentrations within cells. Moreover, again analogous to real-life traffic jams, traffic-jam-like behaviour in plant nutrient uptake has undesirable implications in the biological context as well. High intracellular boron is detrimental to plant health as it causes DNA damage and ultimately causes cell death (*Sakamoto et al., 2011-09*). Plants are therefore likely to have evolved mechanisms to avoid high intracellular boron levels to limit boron-induced damage. Lower throughput across the root implies reduced xylem loading and hence a reduction in the boron absorption efficiency, critical for plant growth at low boron conditions.

To better quantify the magnitude of these effects, we performed a simulation screen to evaluate the expected physiological impact of the transporter swiftness in the form of increased boron levels within the cells due to the instabilities, as well reduction in throughput (see Materials and methods for details). We observe increasing maximum levels in boron concentration for slower regulation swiftness (*Figure 5A*), as well as higher fold-changes in boron concentration (*Figure 5B*). This suggests that sufficiently rapid transport regulation is important to reduce the risk of DNA damage when either considering absolute concentrations or the increase over the basal equilibrium boron concentrations. Moreover, under the conditions in which traffic jams and large-amplitude variations occur, we measured a considerable reduction in total throughput through the tissue (*Figure 5C*). In

**Table 1.** Model Parameters.

| | Parameter | Unit | Description | Acceptable range | | Default |
| | | | | Max. | Min. | |
|---|---|---|---|---|---|---|
| BOR | $\epsilon_B$ | - | Time constant for transporter regulation | - | - | 1 |
| | $\alpha_B$ | µm s$^{-2}$ | Production rate of transporter activity | $4.9 \times 10^2$ | $3.7 \times 10^{-9}$ | $2.0 \times 10^{-1}$ |
| | $\xi_B$ | s$^{-1}$ | Basal degradation rate | | $1.6 \times 10^{-6}$ | $7.6 \times 10^{-2}$ |
| | $k_B$ | µM | Boron concentration for half-maximum in Hill's function | 1000 | 1 | 600 |
| | $d_B$ | - | Amplitude of increased degradation rate by boron | 100 | 0 | 50 |
| | $n_B$ | - | Hill's coefficient | - | - | 2 |
| | $c_B$ | µM | Boron concentration at which the flux reaches its half-maximum value | $\infty$ | 500 | 1000 |
| NIP | $\epsilon_N$ | - | Time constant for transporter regulation | - | - | 1 |
| | $\alpha_N$ | µm s$^{-2}$ | Production rate of transporter activity | $4.9 \times 10^2$ | $3.7 \times 10^{-9}$ | $2.0 \times 10^{-1}$ |
| | $\xi_N$ | s$^{-1}$ | Basal degradation rate | | $1.6 \times 10^{-6}$ | $7.6 \times 10^{-2}$ |
| | $k_N$ | µM | Boron concentration for half-maximum in Hill's function | 1000 | 1 | 20 |
| | $n_N$ | - | Hill's coefficient | - | - | 2 |
| Cell size | lc | µm | Cell width | 20 | 5 | 10 |
| | lw | µm | Cell wall width | 2 | 0.2 | 0.5 |
| | hc | µm | Cell height | 150 | 5 | 20 |
| Other | p | µm s$^{-1}$ | Membrane background permeability of boron | $8 \times 10^{-2}$ | $2.3 \times 10^{-3}$ | $3 \times 10^{-2}$ |
| | a | µm s$^{-1}$ | Xylem loading rate (in the last cell) | 500 | 0 | 0.5 |
| | $c_0$ | µm | Boron concentration in medium | 5000 | 0 | 300 |

DOI: https://doi.org/10.7554/eLife.27038.020

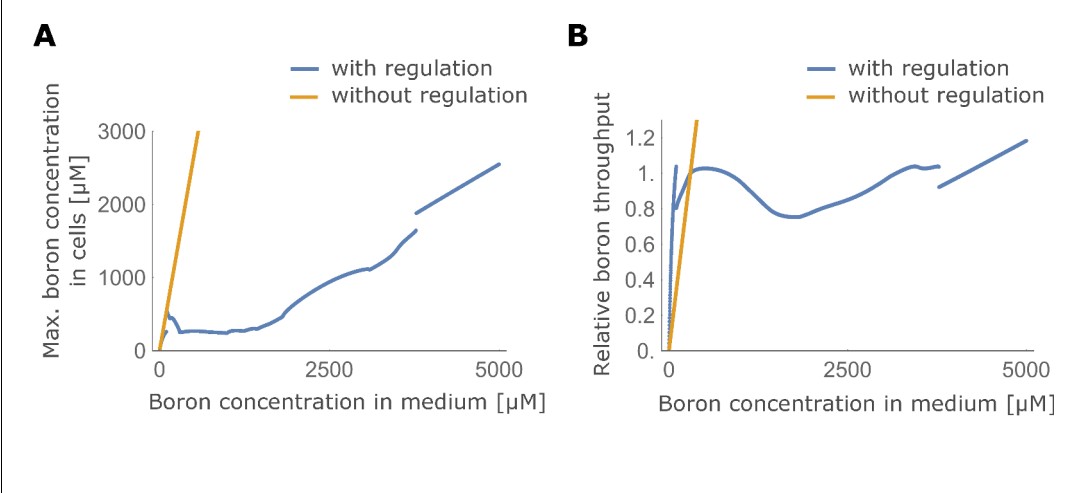

**Figure 8.** Maximum boron concentration and throughput with and without transporter regulation. (**A**) Maximum boron concentration; and (**B**) throughput, for the four-cell NIP-BOR model, plotted for varying boron concentrations in the medium. The default parameters are used (*Table 1*). When transporter regulation is not considered, NIP and BOR activity were fixed to their unregulated equilibrium value in the wild type model, namely $\frac{\alpha_N}{\xi_N}$ and $\frac{\alpha_B}{\xi_B}$.

DOI: https://doi.org/10.7554/eLife.27038.019

our default model we assume the BOR transporter-driven flux to be linear with the cellular boron concentration. If we instead consider Michaelis-Menten saturated BOR transport (using a biologically reasonable Michaelis constant — the boron concentration at which the flux becomes half as large due to saturation, as well as the concentration at which the flux reaches its half-maximum value — of $c_B = 1000$ µM) the system's capacity of uptake and throughput only marginally decreases. In stark contrast, the detrimental effects of the instabilities that arise due to slowing down the regulatory dynamics are amplified when BOR saturation is considered (see details of saturation implementation in the caption of *Figure 4—figure supplement 3*). The impact of the traffic jams becomes much more severe, with peak levels during the concentration fluctuations more than twice as high as the default scenario (compare *Figure 2* to *Figure 4—figure supplement 3*). This is a direct consequence of the cell not being able to

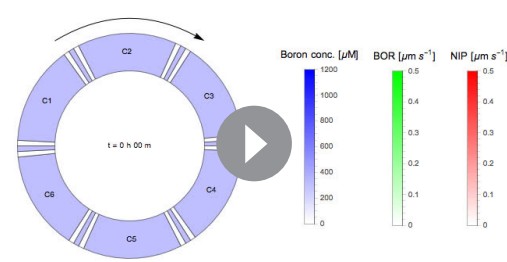

**Video 1.** Simulation of six identical polarised cells endowed with BORs and NIPs orientated as to generate clockwise boron flows. Time is indicated above the simulation, and dynamics show an initially constant flow, with low boron concentrations in all cells. Due to the intrinsic instability of the system, minute numerical noise generates boron peaks in the cells, a pattern which propagates counterclockwise (i.e., against the flow) through the tissue. Cytosolic boron concentrations, as well as those in the cell walls between cells, are depicted through the colour bar shown in the movie. Transporter levels (of both BORs and NIPs) are characterised through the colour changes along the radial membranes, again depicted through the colour bars shown in the movie.
DOI: https://doi.org/10.7554/eLife.27038.016

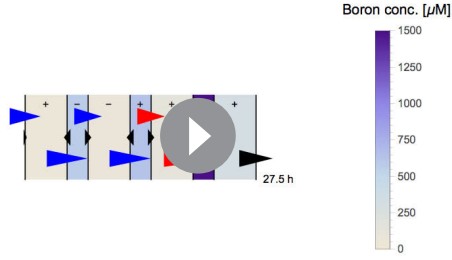

**Video 2.** Dynamics of transporters and fluxes. Simulation output revealing the details of boron flows over the membranes, due to transporter dynamics and background membrane permeability rates.
DOI: https://doi.org/10.7554/eLife.27038.023

**Table 2.** Probes used in the smFISH experiments.

| NIP5;1 exon probes | NIP5;1 intron 1 probes | PP2A mRNA |
|---|---|---|
| acgaaaatggagctaggact | gtagtcgattttcttacggt | ccgagcgatctatcaatcag |
| cggtttcaccaaacacaagt | attactagcacaaaccactt | gacatcctcaccaaaactca |
| gtgttttaaacttcgccagt | cggatggtgacgaatgagta | tcgggtataaaggctcatca |
| tggaggagccatcaccatta | atgcaatatgcggttatgga | tagctcgtcgataagcacag |
| cattgaatccactctcatcc | tctattgttaggtttactga | ccaagagcacgagcaatgat |
| gttggtttccgatgatcaaa | ggttcctagccagaaatttt | atcaactcttttcttgtcct |
| cggcaagcatttgcatcgag | tgtaatttttaggcttacgt | catcgtcattgttctcacta |
| atgttgaccccaagttgatc | ctttgtaaggttataacgct | atagccaaaagcacctcatc |
| gagggaaaatcggtgaagca | aacgagccattggatttctt | atacagaataaaaccccccca |
| ttgcgagtgagggagacatc | cgtggattccaatgttttct | caagtttcctcaacagtgga |
| tgaatgttcccacgaactcg | ttcccaattcattattttcc | tcatctgagcaccaattcta |
| gctgtcgcggtgaatatcaa | tgtctcgatctcattttttt | tagccagaggagtgaaatgc |
| catcgtatttctggttcacg | ccctatagtacatctatatt | cattcaccagctgaaagtcg |
| gttaccgattagggtttctg | attacgatcgatttgtgagt | ggaaaatcccacatgctgat |
| tatgatcatcactgcgagtc | ttctctgtatttcagagctt | atattgatcttagctccgtc |
| cctgagatatgacctgttga | agtattatcttttggtcact | attggcatgtcatcttgaca |
| cagtgatgggtttaggtgag | ccctatgatcttttcaata | aaattagttgctgcagctct |
| cttagagcagcgaatgctat | ctcctccaagtgtgacgtaa | gctgattcaattgtagcagc |
| tatgtaagcagggacgtgtg | ctagttcatgtcgtgttaaa | ccgaatcttgatcatcttgc |
| gcgcaaatggaagctgagac | atttaattctcggttgcgac | caaccctcaacagccaataa |
| gaacactcctttaagtgcga | ttcactctcttttctatttgc | ctccaacaatttcccaagag |
| caccaccggacataaaagga | ctcttagtttttcttagact | caaccatataacgcacacgc |
| tccaaggctaacagatggaa | taagttgagatcgagtgggc | agtagacgagcatatgcagg |
| tgaactcaagagcaaaggct | gagtcggtgtccattgaata | gaacttctgcctcattatca |
| cggcagttacaacaaagagg | actattagctcattgtcaca | cacagggaagaatgtgctgg |
| aacggcacgagtgtcggtag | atttacgcaacttgcgtgtt | tgacgtgctgagaagagtct |
| aacggctatacctgccaatt | agcgtgaccgtttattttt | cccattataactgatgccaa |
| aatattgagcatgaccgtgg | actttgtattcgtcattgca | tggttcacttggtcaagttt |
| atagatccaccagtcgatgg | gacctaaccaaaccatacgt | tctacaatggctggcagtaa |
| tcctagagttctcacaggat | gcttccgtcatggacagaaa | cgattatagccagacgtact |
| atagtttcctgatgcaacgg | ctaaaagaatcccatccggt | gactggccaacaagggaata |
| ccagatacacccatagtgac | aaaggacaagagccgtggat | catcaaagaagcctacacct |
| cagatatggcaccaagtgta | tccacgttaacgagcatcaa | ttgcatgcaaagagcaccaa |
| ttaacacctgtgtagaccgc | gcgtgtgatctgtctatatc | acggattgagtgaaccttgt |
| gtcagtcacgctatcgttaa | tacagtcaacggttagtatt | cttcagattgtttgcagcag |
| aaagctcctaaccggacgag | gtcattagtagttactagtt | ggaccaaactcttcagcaag |
| gtctctcactcacttaacga | agtattatttctgctgtcca | ggaactatatgctgcattgc |
| tccaaagctttttcatatca | ctagagggttctgagtcgaa | gtgggttgttaatcatctct |
| gccttttattcttcacacaa | gagatgtttcttgtctaaca | tgcacgaagaatcgtcatcc |
| tgcgtggacttatagtcaca | gtactagatgaatagaggct | ttactggagcgagaagcgat |
| aggtttatatagaccgatgc | | gaacatgtgatctcggatcc |
| aatgacgagacaagttcgca | | ctctgtctttagatgcagtt |
| agaaacccaaaccacacata | | catcattttggccacgttaa |
| ataaaaacagcctcgtctcc | | cgtatcatgttctccacaac |

*Table 2 continued on next page*

*Table 2 continued*

| NIP5;1 exon probes | NIP5;1 intron 1 probes | PP2A mRNA |
|---|---|---|
| acacatgccatagattttat | | atcaacatctgggtcttcac |
| | | ttggagagcttgatttgcga |
| | | acacaattcgttgctgtctt |
| | | cgcccaacgaacaaatcaca |

DOI: https://doi.org/10.7554/eLife.27038.021

effectively efflux boron when intracellular levels spike. Taken all into consideration, we infer that selective pressures operating on the root's capacity to absorb boron optimally and robustly could result in the system evolving to a regime of rapid regulation of transporters.

## Traffic jam-like behaviour is independent of bottlenecks in the tissue context

Our model thus far consisted of a segment of polarised tissue, of a variable number of cell files, linking the soil to the xylem via a polar flow of nutrients. Consequently, both soil and xylem, although presenting stable characteristics over time, do constitute abrupt boundary conditions: the first sets a constant medium concentration, and the second presents a constant convective flow away from the transversal section. We thus questioned if these boundary conditions are responsible for triggering the traffic jam-like regime that arises under lower rates of transport gene regulation. During the last decades a similar discussion has been taking place regarding what triggers the analogous traffic jams in discrete macroscopic systems such as road networks (*Kesting and Treiber, 2013*). Although traffic jams are readily triggered by bottlenecks (such as road obstructions, roundabouts, etc.), theoretical models of traffic jam behaviour predicted that the collective systems parameters alone could be sufficient to render the free flow state unstable. It depends on the car density exceeding a critical value as well as how speed relates to the local car density. To prove this theoretical insight, *Sugiyama et al., 2008* developed an experiment in which cars were confined to move on a homogeneous circular road. Above a critical car density they observed a transition from a free flow to a traffic jam state, due to the collective effect of the vehicles. They concluded that, although no bottleneck was present, the intrinsic parameters of the system rendered it unstable, such that the smallest of naturally occurring fluctuations was enhanced to disrupt the free flow. This experiment illustrates the non-intuitive notion of traffic jams being generated spontaneously under certain critical parameter regimes.

Inspired by this strategy, and for us to be able to rule out that the origin of the nutrient-flow instabilities within the plant tissue is provided by the soil boron concentration or the xylem flow, we constructed an in silico polarised tissue composed of 6 cells, wrapped up into a ring, that is, the first cell is connected to the last. The ring structure avoids boundary conditions (i.e., bottlenecks) and offers a scenario in which uninterrupted flows (either stable or unstable) can be studied. We allow the dynamics of the polarised tissue to evolve from an initial situation in which concentrations are homogeneous over the whole ring-tissue. When transporter regulation is as swift as in wild type (*Figure 6A*), a steady flow with no oscillation in the concentrations is observed. When transporter regulation swiftness is decreased to sufficiently low values as to cause unstable flows (as derived from the analysis shown in *Figure 4*), oscillations arise. After an initial period in which the flows seem constant, random numerical fluctuations in the simulation bring forth a rise of boron concentration in a random cell, which, due to the mechanism of inhibition by boron of the boron transporters, results in local interruption of flows and the back propagation of the boron concentration pulse (*Figure 6C, D*, *Video 1*). The frequency of these oscillations is decreased, and the wave broadened, as the regulation speed is decreased (*Figure 6C,D*). These simulations demonstrate that the unstable flow regime — which we now can conclusively eliminate as being triggered by boundary conditions — continuously self-propagates.

**Table 3.** Primers used in qRT-PCR experiments.

| Name | Sequence | Description |
|------|----------|-------------|
| NIP5;1_qRT-PCR-1 | caccgattttccctctcctgat | Probes for realtime PCR |
| NIP5;1_qRT-PCR-2 | ctctttcttactctctagcctc | |
| NIP5;1_qRT-PCR-3 | gaatgttcccacgaactcgg | |
| NIP5;1_qRT-PCR-4 | acattcatcttgatattcacc | |
| NIP5;1_qRT-PCR-5 | gcatgcagcgttaccgatta | |
| PP2A_qRT-PCR-1 | ccatgtttgaggatcttacgc | |
| PP2A_qRT-PCR-2 | tgtctacatctcagcttcagtgtc | |
| PP2A_qRT-PCR-3 | gctccaacaatttcccaaga | |
| PP2A_qRT-PCR-4 | caagattcggttagattattg | |
| PP2A_qRT-PCR-5 | gcgagaaattgacaatcacag | |
| NIP5;1_mRNA_F | atttaggtgacactatagtaagctcaaagactaaccaaac | For amplification of the standard DNA fragments |
| NIP5;1_mRNA_R | ttacacatgccatagattttat | |
| PP2A_mRNA_F | atttaggtgacactatagcggtctcatttctcgttcttc | |
| PP2A_mRNA_R | cacttgataagtaaattatttg | |

DOI: https://doi.org/10.7554/eLife.27038.022

## Mechanisms underlying rapid regulation

Our model analysis highlights the requirement for rapid regulation of both BOR1 and NIP5;1 transporters to stabilise boron flux through root cells and minimise risks associated with transient high levels of boron in cells, both considered to be important constraints for the plant. Current experimental evidence shows rapid downregulation of the BOR1 protein (*Takano et al., 2005*; *Takano et al., 2010*), and *NIP5;1* transcript (*Tanaka et al., 2011*; *Tanaka et al., 2016*), although the fine details regarding the molecular mechanisms remain to be elucidated. The modelling however predicts permanently fast regulation, not only when boron levels drop, but also when they remain constant. This is however difficult to prove with techniques that we have used previously.

To experimentally explore the regulatory swiftness of the bidirectional transporter NIP5;1 at the transcript level, we used single molecule RNA FISH (smFISH). In this method ~40 complementary fluorescently labeled oligonucleotide probes are used to visualise RNA at the cellular level (*Raj and Tyagi, 2010*; *Duncan et al., 2017*). Initially, we combined probe sets to detect mRNA for both *NIP5;1* and the housekeeping gene *PP2A* (specifically A3 scaffolding subunit of PP2A, At1g13320) and compared abundance and cellular distribution (*Figure 7A*) (*Czechowski et al., 2005*; *Duncan et al., 2016*). Consistent with published smFISH experiments, *PP2A* mRNA appeared evenly diffused throughout all cells (*Figure 7B*) (*Duncan et al., 2016*). In contrast, we observed many nuclear accumulations of *NIP5;1* RNA that were accompanied by few cytoplasmic mRNA copies (125 nuclear accumulations in 149 cells) (*Figure 7B*). Large nuclear smFISH RNA signals have been reported for highly transcribed genes. They are typically accompanied by many copies of cytoplasmic mRNA and considered to represent bursts of transcription, where multiple Pol II associate with a locus (*Battich et al., 2015*; *Duncan et al., 2016*).

Arabidopsis genes that contain introns must undergo pre-mRNA splicing before mature transcripts are translated (*Morello and Breviario, 2008-06*). Co-transcriptional splicing is common for plant genes and this ensures that introns are removed from the pre-mRNA and rapidly degraded close to the site of transcription (*Reddy et al., 2013-10*). This system allows intronic smFISH RNA labelling to be used as an effective method to identify loci actively engaged in transcription (*Levesque and Raj, 2013*; *Duncan et al., 2016*; *Rosa et al., 2016*). We used this approach to further investigate *NIP5;1* transcription and found 72% of cells with at least one *NIP5;1* intron signal. As expected, this was lower than the 84% previously reported for the more highly expressed gene *PP2A* (*Duncan et al., 2016*). When we combined *NIP5;1* exonic and intronic RNA smFISH probe sets we observed 92.5% of nuclear accumulations co-localised with intron signals ($n = 111$) (*Figure 7C*).

This is consistent with the nuclear accumulations representing sites of ongoing RNA production and degradation, rather than separate nuclear storage compartments.

The suggested rapid degradation of *NIP5;1* mRNA after transcription was further supported by an independent approach, in which we performed qRT-PCR using probes specific to pre- or mature mRNA to measure mature mRNA/pre-mRNA ratio in root cells. The average mature mRNA/pre-mRNA ratio of *NIP5;1* was less than 25% of that of *PP2A* (***Figure 7—figure supplement 1***). In accordance with our interpretation of the smFISH results, this again indicates a high degradation rate of *NIP5;1* mRNA.

In light of our model predictions, these observations combined point to a highly dynamic sensing system where RNA is turned over at or near the site of transcription, to limit *NIP5;1* levels when boron is above a threshold level. We recently demonstrated that *NIP5;1* transcript degradation occurs through ribosome stalling, triggering degradation in the cytoplasm (***Tanaka et al., 2016***). We also identified an upstream sequence that promotes mRNA degradation, but does not trigger ribosome stalling. The data presented in ***Figure 7*** provides evidence of mRNA degradation occurring close to the site of transcription. Together these results suggest that mRNA degradation could occur through two coordinated B-dependent processes; one where cytoplasmic degradation is triggered by ribosome stalling and another localised in the nucleus where RNA is turned over at the site of transcription. Relief of this repression could then ensure rapid boron uptake when required. Our model suggests that such a dynamic mRNA production/decay process could support the necessary swift boron-mediated transporter regulation critical for the evolution of controlled, stable boric acid flows through polarised tissue.

These results also support the notion of a constant turnover (both producing and breakdown the RNA) occurring for regulating *NIP5;1* transcript levels. We propose that this 'wasteful' process can now be understood in light of constraints of stable flows through polarised tissue. Interestingly, the regulation of the transporters is of a different nature, one predominantly on the level of protein degradation (BOR1), the other occurring on the level of transcript accumulation (NIP5;1).

## Conclusion

We conclude that the experimentally observed swiftness of boron transporters' boron-mediated downregulation can be understood as a necessary mechanism to maintain optimal constant xylem loading rates over time, while also avoiding DNA damage due to oscillations within cells. One could argue that without a substrate-mediated downregulation of the transporters there would not be the need to avoid traffic jam-like behaviour through swift regulatory dynamics in the first place. However, were the system not to present the regulation, the plant would be unable to shield itself from toxic soil boron levels (***Figure 8A***) and be much less efficient in growing at low soil boron levels (***Figure 8B***). Thus, some kind of inhibitory regulation necessarily has to be present, to ensure plant plasticity under different environmental conditions. As a consequence, the speed of the transporter downregulation needs to be sufficiently high. We showed that this is an intrinsic requirement, simply because boron transporters are polarly located: The requirement itself does not depend on external boron fluctuations in the medium, number of cell files composing the root tissue, possible saturable activity of the transporter itself nor the magnitude, location or strength of the xylem flow.

Our study has been based on the well-characterised system of directed boron transport, owing to the richness in quantitative experimental measurements regarding BOR and NIP regulation. However, the implications of our derived insights apply to any system in which the dynamics of polar transport of a given substance relies on an inhibitory feedback between the concentration of that substance and its local transport rate. For example, the phosphate and iron transport systems present both polar transporters as well as substrate-dependent regulation, serving as candidates for such phenomena and dynamical constraint avoidance. In the case of the phosphate transport system, transporters localised to the outer side of the epidermis are responsible for uptake (LePT1 and LePT2, ***Liu et al., 1998***), and they have recently been found to carry a domain possibly involved in phosphate sensing which leads to rapid degradation of phosphate transporter in a phosphate-dependent manner (***Gu et al., 2016***). Similar regulation is also reported for the iron (Fe) transport system, where polar accumulation of IRT1, a major Fe transporter for Fe uptake into roots, is regulated in an Fe-dependent manner (***Barberon et al., 2014***). Even other transport systems which do not shuttle nutrients might display this sort of instability, such as polar auxin transport, for which stable flows might also require fast response dynamics regarding the activity of auxin importers and

efflux facilitators (*Robert and Friml, 2009*). Indeed, for any polarised tissue in which adaptive regulation of transport levels is necessary we can expect to find intrinsic temporal constraints operating to ensure that steady state flows are maintained and large fluctuations avoided.

In molecular terms, we discovered that although both transporters are predicted to work synergistically and are both required to be effectively similar in regard to rapid regulation, their regulations are biologically encoded in unique ways. We here experimentally focused on the regulatory swiftness of the bidirectional transporter NIP5;1, and found strong evidence that it constitutes a system in which production is maintained high but degradation is rapidly controlled transcriptionally, allowing for the necessarily rapid boron-dependent response. It will be interesting to speculate what underlying reason, if any, has caused the directed exporter, BOR1, which is similarly known and required to be swiftly regulated, to be controlled on a biologically distinct level.

For this work, the analogy with motorway traffic jams was helpful to build an understanding of principles of polarised transport dynamics, albeit both systems obviously deviate in important aspects. Boron concentrations are continuous and can reach arbitrarily high levels, as opposed to (incompressible) cars. Yet, the discrete nature of cells and their regulated polar transporters, combined with the boron-dependent inhibition of the transporters themselves, resembles cars slowing down in response to busy regions along the motorway. Moreover, in both systems highly sensitive responses can prevent jamming (*Sugiyama et al., 2008*). We therefore propose that the notion of traffic jams can be universally instructive and helpful for biologists considering physiology and regulated growth through polarised transport in plants, as well as for future efforts to enhance plant growth under diverse environmental conditions.

## materials and methods

### Observation of protein localisation of boron transporters

For plant culture, MGRL medium was used (*Fujiwara et al., 1992*) supplemented with 1% sucrose and solidified with 1.5% gellan gum. For *Figure 1*, *A. thaliana* transgenic lines carrying *BOR1-GFP* (*Kasai et al., 2011*) and *GFP-NIP5;1* #8 (*Tanaka et al., 2011*) were germinated and pre-incubated for 5 days on normal MGRL medium plate, and then grown for 2 days on MGRL plates with 0.3 μM boric acid. For *Figure 1—figure supplement 2*, plants were grown for seven days on MGRL medium with the indicated boric acid concentrations. Images were captured with a confocal microscope (FV1000, Olympus, Japan). GFP fluorescence was observed with 473 nm excitation and 510 nm emission. Cell wall was stained with 10 μg/mL propidium iodide aqueous solution for 3 min, and observed with 559 nm excitation and 619 nm emission.

*NIP5;1* promoter activity was observed using a transgenic *A. thaliana* strain carrying *NIP5;1* promoter-GUS (strain −2,180ΔUTR312 #8 in *Tanaka et al., 2011*). Seedlings were germinated and cultured on MGRL medium plate. Four-day-old seedlings were stained with GUS staining solution: 100 mM $Na_2HPO_4$ (pH 7.0), 0.1% Triton X-100, 2 mM $K_4Fe(CN)_6$, 2 mM $K_3Fe(CN)_6$, 0.5 mg/mL X-Gluc (5-bromo-4-chloro-3-indolyl β-D-glucuronide cyclohexylammonium Salt, Glycosynth, UK) in a decompressed desiccator for 20 min at room temperature. After rinsed with phosphate buffer (pH 7.0), stained seedlings were mounted in 45°C molten 5% agar (gelling temperature 30–31°C, Nacalai, Japan) aqueous solution and solidified at 4°C for 3 hr. The samples were trimmed into blocks and sliced with a vibrating microtome with thickness of 100 μm. The sliced sections were observed with microscope.

### Model parameter choices

Although we have constructed our model as parsimonious as possible to qualitatively explore what kind of behaviours could emerge from a polarised tissue with transporter regulation, our model requires a few important parameters to be set. We have sought to explore the possible dynamics of our system while staying well within the expected limits of what is experimentally considered plausible in terms of the boron transport system. Below we give a brief description of the data our parameter choices were based on, and, when possible, how reasonable ranges could be established.

The plasmamembrane boron permeability rate ($p$) is set to a maximum rate of $8 \cdot 10^{-2}$ $s^{-1}$ and a minimum rate of $4.4 \cdot 10^{-3}$ $s^{-1}$, based on the estimation of boric acid membrane permeability by *Raven, 1980* and on more recent experimental measurements performed on internodal cells of the

charophyte alga *Chara corallina* (*Stangoulis et al., 2001*). Note that the plasmamembrane's permeability rate for boric acid is relatively high compared to other nutrient elements, due to boric acid existing in a non-charged form under physiological pH.

The degradation rates $\xi_B$, $\xi_N$ represent the effective decay rate of the actual transporter activity, not just of the transporter protein. This includes processes such as its removal from the membrane, its inactivation due to usage, etc. As it amalgamates a range of inactivation/degradation processes, we set it higher than typical protein degradation rates, but lower than typical membrane residence times, using a half life of 9 s. The valid ranges for the production rate of the transporter activities, $\alpha_B$ and $\alpha_N$, were derived from the degradation rates $\xi_B$, $\xi_N$, combined with qualitative conditions for the transporter equilibrium levels. It is as yet not possible to obtain these values directly through experiments, as several intermediate steps are involved. Instead we impose that the transporter activity steady state level does not exceed the plasmamembrane's permeability by a thousand fold:

$$p < B^*, \quad N^* < 1000p. \tag{8}$$

Combining these conditions we could then extract the valid ranges for $\alpha_B$ and $\alpha_N$. As explained in the main text, we take $k_B \simeq 30k_N$, based on our observations of the sensitivity of BOR and NIP5;1 expression under different boron concentrations (*Figure 1—figure supplement 2*), further corroborated by earlier data on NIP5;1, as displayed in Figure 1B in *Tanaka et al., 2011*. We also ran simulations to show robustness of the observed behaviours for smaller variations between $k_B$ and $k_N$ by assuming lower values of $k_B$ (*Figure 2—figure supplement 1*, *Figure 4—figure supplement 2*). The parameter $d_B$ determines the maximum fold increase in BOR degradation at very high intracellular boron concentrations. It was set to a 50-fold increase, based on rough assessments of experiments in which plants were transferred to high boron conditions, with the caveat that assessment of the temporal change in functional, membrane-located BOR as a function of the intracellular boron concentration is very challenging. Typical cell sizes (height, $hc$; width, $lc$) and cell wall width ($lw$) were estimated from confocal microscope images of *A. thaliana* roots. The xylem loading rate $a$ was calculated as follows: *Hosy et al., 2003* reported that in mature Arabidopsis the rate of transpiration is 0.1 g water/g FW tissue/hour at 70% humidity under daytime conditions. Given an Arabidopsis FW at 18 days of around 0.2 g/plant, transpiration is 20 mg/hour/plant, or roughly $5 \cdot 10^6$ $\mu m^3$/s. Considering a stem xylem cross-sectional surface area being roughly $10^7$ $\mu m^2$, the convective velocity $a$ should therefore be around $5 \cdot 10^6$ $\mu m^3$/s / $10^7$ $\mu m^2$ = 0.5 $\mu m$/s.

## Numerical analysis

Numerical and analytical calculations were computed using Wolfram Mathematica 10.2. To calculate time development of the model, ODEs were solved numerically using the NDSolve[] function, setting the initial conditions for each variable to zero. The parameters used in the models are shown in *Table 1*. The solutions of the ODEs were visualised as a line plot or kymograph, using the Plot[] or ListDensityPlot[] functions.

To evaluate the physiological impact (maximum concentration and amplitude of cytosolic boron, and throughput towards shoots) of transporter regulation swiftness, numerical simulations were repeated for combinations of $\epsilon_B$ and $\epsilon_N$, which were independently varied between 0.01 and 2, at intervals of 0.01. For each simulation, minimum and maximum boron concentration for each cell and average throughput through the cell row between 24 to 48 h simulation time were recorded using the 'EventLocator' method of the NDSolve[] function.

For the ring model, the boundary conditions in the original model were removed by connecting the last cell in the tissue-segment to the first one. This was done by using a tissue composed of six cells and then introducing a cell wall between cell 6 and cell 1. The boron concentration in the novel cell wall is referred to as $W_{6-1}$. This generates a six-cell ring model. Because within a ring-model the total amount of boron is conserved, introducing an additional ODE for this cell wall component effectively means an overdetermination of variables, causing numerical issues. Hence the value of the new variable was instead defined as follows:

$$W_{6-1} = \left( B_{total} - lc\,hc \sum_{i=1}^{6} C_i - lw\,hc \sum_{i=1}^{5} W_i \right) \frac{1}{lw\,hc}, \tag{9}$$

where $B_{total}$ is the total amount of boron in the whole system, which is constant over time as there is no input into nor outflow from the system. The initial boron concentration of all cells and cell walls in the simulation were set to 300 μM, and $B_{total}$ was determined accordingly.

The local stability analysis was performed as follows: First, all possible equilibria for the set of ODEs were found using the NSolve[] function. Next, analytical descriptions of the Jacobian matrix of the ODEs around the equilibria were derived using the D[] function. Finally, the stability of the system for varying values of $\epsilon_B$ and $\epsilon_N$ was determined according to the signs of the eigenvalues of the Jacobian matrix. (Please note that changing $\epsilon_B$ and/or $\epsilon_N$ does not change the number of equilibria or their value, only their stability can change.) The system is stable when the real parts of all the eigenvalues are negative, and unstable in any other case. The imaginary part of the largest eigenvalues was then used to estimate the oscillatory period in the unstable region, based on the following conversion:

$$Oscillatory\,period = \frac{2\pi}{\lambda_{im}} \,. \tag{10}$$

To compare the predicted oscillatory period close to the unstable equilibrium with the actual oscillatory period of the system, the oscillatory period was also calculated numerically (*Figure 4—figure supplement 1*). For certain parameters individual variables can present rather complicated patterns, for example temporarily maximum values, followed by a slight decrease and then further rise before decreasing again. We therefore used the following algorithm to determine the oscillatory period: The ODEs were numerically solved using the NDSolve[] function. The 'EventLocator' method was then used to collect all extrema for three variables, $C_2$, $C_3$ and $N_2$, for the period between 24 and 72 hr. Using the first set of extrema as a reference point, the second and potentially later sets of extrema were then compared against the first set. The first case for which all differences were within ±0.1% for all three variables was then labeled as being identical to the reference point, and the corresponding time interval reported as the oscillatory period.

## smFISH probe design

The probes were designed using the online program Stellaris™ Probe Designer version 2.0 and ordered from LGC Biosearch Technologies (Petaluma, CA). For probe sequences see *Table 2*.

## Observation of *NIP5;1* RNA

Col-0 seeds were stratified for 2 d at 4°C and then germinated and grown under 16/8 hr light conditions at 20°C on MGRL plates (30μM boric acid). smFISH was carried out on seedlings as described by *Duncan et al., 2017*. Briefly, 5d old seedlings were fixed for 30 min in 4% paraformaldehyde. Root squash samples were prepared on slides. Tissue permeabilisation was achieved by immersing the samples in 70% ethanol for a minimum of one hour. Two washes were carried out with wash buffer (10% formamide and 2x SSC). 100 mL of hybridisation solution (0.3 M Sodium chloride and 30 mM tri-Sodium citrate dihydrate, referred to as '2x SSC', 10% dextran sulfate and 10% formamide) containing probes (final concentration 25 nM) was then left to hybridise at 37°C in the dark overnight. Each sample was washed twice with wash buffer following hybridisation buffer removal with the second wash left to incubate for 30 min at 37°C. Each slide was then incubated with 100mL 4′ 6-diamidino-2-phenylindole dihydrochloride (DAPI, Sigma, St. Luis, MO) (100 ng/mL) at 37°C for 30 min. The DAPI solution was removed before 100 μL 2x SSC was added and then removed. Samples were then incubated for 2 min at room temperature with 100 μL GLOX buffer (0.4% glucose in 10 mM Tris, 2x SSC) before being mounted in 100 μLof GLOX buffer containing 1 μL of glucose oxidase (#G0543, Sigma) and 1 μL catalase (Sigma) under 22 mm x 22 mm No. 1 coverslips (Slaughter, Uppminster, UK) and sealed by nail varnish. A Zeiss Elyra PS1 inverted microscope was used for imaging. A x100 oil-immersion objective (1.46 NA) and cooled EM-CCD Andor iXon 897 camera (512×512 QE>90%) were used to obtain all images. (Quasar®570 probes: excitation by 561 nm laser, signals detected between 570–640 nm. Quasar®670 probes: excitation by 642 nm laser, signals detected between 655–710 nm. DAPI: excitation by 405 nm laser, signals detected between 420–480 nm.) For all experiments a series of optical sections were set up with z-steps of 0.2 μm. All smFISH images were de convolved with Huygens Professional version 16.05 (Scientific Volume Imaging, the Netherlands, http://svi.nl) using the QMLE algorithm with Good signal to noise settings and

50 iterations. Maximum Z projections in *Figure 7* were performed using Fiji (an implementation of ImageJ, a public domain program by W. Rasband available from http://rsb.info.nih.gov/ij/).

## Determination of mature mRNA/pre-mRNA abundance ratio

Col-0 seedlings were germinated and grown on MGRL plates containing 30 µM boric acid for eight days. Total RNA was extracted from whole roots of about 25 seedlings with four replicates, using NucleoSpin RNA Plant (MACHEREY-NAGEL). To reduce contamination of genome DNA, 500 ng of prepared RNA was subjected to DNase treatment with RQ1 RNase-Free DNase (Promega), following the manufacture's protocol. Reverse transcription was performed with Super Script III reverse transcriptase (Thermo Fisher Scientific) with random primers. To compare the abundance of cDNAs with different sequences, absolute quantification by realtime PCR was conducted, using PCR-synthesised DNA fragments with known concentration as standard templates. To detect mature or pre-mRNA specifically, exon-exon or intron-exon probes were designed (*Figure 7—figure supplement 1* and *Table 3*). Realtime PCR was performed with Thermal Cycler Dice® Real Time System and SYBR Premix Ex Taq II (Tli RNaseH Plus) (Takara), following the supplied standard shuttle 2-step PCR protocol. As standard DNA templates, DNA fragments corresponding to cDNA of mature- or pre-mRNA of NIP5;1 and PP2A were amplified by PCR with Prime Star GXL (Takara), using Col-0 cDNA or genome DNA as a template. The primers used are shown in *Table 3*. The PCR products were subjected to electrophoresis and the target fragment was purified from the gel slices with QIAquick Gel Extraction Kit (Qiagen). The concentration of the purified DNA was estimated by NanoDrop 1000 (Thermo Fisher Scientific). Tenth dilution series of 1 pg/µL~1 ag/µL template DNA were applied for standard curve. The copy number of each standard DNA solution was normalized by realtime PCR $C_t$ values of the exon-exon proves which amplify both mature- and pre-mRNA sequences (*Figure 7—figure supplement 1* and *Table 3*). As a negative control, in parallel, after the total RNA extraction, the same samples were processed with the same procedure except reverse transcription, confirming that realtime PCR signals from contaminated genome DNA does not affect the final results.

## Acknowledgements

This work has been supported by the UK Biological and Biotechnology Research Council (BBSRC) via grant BB/J004553/1 to the John Innes Centre and by the Japan Society for the Promotion of Science via Grants-in-Aid for Scientific Research (No. 15J11021 to NS and No. 25221202 to TF). Experiments were partly financed by the OpenPlant BBSRC/EPSCR Synthetic Biology Research Centre (Grant BB/L014130/1).

## Additional information

### Funding

| Funder | Grant reference number | Author |
| --- | --- | --- |
| Biotechnology and Biological Sciences Research Council | BB/J004553/1 | Athanasius FM Marée Verônica A Grieneisen |
| Japan Society for the Promotion of Science | 25221202 | Toru Fujiwara |
| Engineering and Physical Sciences Research Council | BB/ L014130/1 | Susan Duncan |
| Japan Society for the Promotion of Science | 15J11021 | Naoyuki Sotta |

The funders had no role in study design, data collection and interpretation, or the decision to submit the work for publication.

### Author contributions

Naoyuki Sotta, Data curation, Software, Formal analysis, Funding acquisition, Validation, Investigation, Visualization, Writing—review and editing; Susan Duncan, Data curation, Funding acquisition, Validation, Investigation, Methodology, Writing—review and editing; Mayuki Tanaka, Validation,

Investigation; Takafumi Sato, Formal analysis, Investigation; Athanasius FM Marée, Conceptualization, Software, Formal analysis, Supervision, Investigation, Methodology, Writing—original draft, Project administration, Writing—review and editing; Toru Fujiwara, Conceptualization, Resources, Formal analysis, Supervision, Funding acquisition, Investigation, Methodology, Project administration, Writing—review and editing; Verônica A Grieneisen, Conceptualization, Resources, Formal analysis, Supervision, Funding acquisition, Investigation, Methodology, Writing-original draft, Project administration, Writing-review and editing

## Author ORCIDs
Naoyuki Sotta (iD) https://orcid.org/0000-0001-5558-5155
Susan Duncan (iD) https://orcid.org/0000-0001-9581-1145
Athanasius FM Marée (iD) https://orcid.org/0000-0003-2689-2484
Toru Fujiwara (iD) http://orcid.org/0000-0002-5363-6040
Verônica A Grieneisen (iD) https://orcid.org/0000-0001-6780-8301

## Decision letter and Author response
Decision letter https://doi.org/10.7554/eLife.27038.027
Author response https://doi.org/10.7554/eLife.27038.028

# Additional files

## Major datasets
The following dataset was generated:

| Author(s) | Year | Dataset title | Dataset URL | Database, license, and accessibility information |
|---|---|---|---|---|
| Naoyuki Sotta, Susan Duncan, Mayuki Tanaka, Takafumi Sato, Athanasius FM Marée, Toru Fujiwara, Verônica A Grieneisen | 2017 | Data from: Rapid transporter regulation prevents substrate flow traffic jams in boron transport | http://dx.doi.org/10.5061/dryad.9pq6k | Available at Dryad Digital Repository under a CC0 Public Domain Dedication |

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
