## [Decision Letter]

Thank you for submitting your article "Rapid transporter regulation prevents substrate flow traffic jams in boron transport" for consideration by *eLife*. Your article has been reviewed by three peer reviewers, one of whom is a member of our Board of Reviewing Editors, and the evaluation has been overseen by Naama Barkai as the Senior Editor. The following individuals involved in review of your submission have agreed to reveal their identity: Christophe Maurel (Reviewer #2).

The reviewers have discussed the reviews with one another and the Reviewing Editor has drafted this decision to help you prepare a revised submission.

Summary:

Previously, co-authors of this manuscript described the coordinated, polar locations of the NIP5 and BOR1 transporters and the surprising finding of their rapid boron-dependent regulation. Here, the authors have used a mathematical modelling approach to explore the significance of this rapid regulation and its necessity. Their model predicts oscillatory patterns of boron accumulation within a cell file and also a spectacular traffic jam behavior when the swiftness of transporter regulations falls below a certain threshold. Thus the authors argue that rapid regulation of transporters is needed in order to ensure unobstructed trafficking across cells and to avoid potentially toxic spikes of intracellular boron. Their findings provide an intriguing new way of thinking about nutrient transport in plants and the concepts they develop are potentially of significance for other polarized transport systems. The reviewers agree on these very positive aspects and also agree that the following points need to be addressed prior to publication.

Essential revisions:

1) The authors hypothesize complex relationships between B concentration and NIP or BOR protein abundance and thus activity (Equation 5). However, they may oversimplify the relationship between B concentration and transport itself. Proportionality of transport to B concentration gradients (Equations 1 and 2) is consistent for channels such as NIPs. Is it always applicable to BOR1, assuming that this transporter may saturate at high *C_i_*? The authors should know the likely *K_m_* of BOR1 (In any case, the *bor1-1* mutant lacks a saturating B uptake component): Under steady state conditions (without oscillation), estimated *C_i_*seems to go up to 250 μM and in oscillating conditions, it goes up to 1.5 mM. In the latter conditions, BOR1 must clearly saturate? In our opinion, the authors should explore this important point.

2) We wondered why the authors hypothesize such a huge difference (30-fold) between *k_B_* and *k_N_*. Is it consistent with actual variations of intracellular B concentration? Data from Figure 1—figure supplement 2 refer to external B concentration and intracellular B may not vary so much. Could the authors present a sensitivity analysis of their model to *k_B_/k_N_*ratio?

3) While the authors have chosen a rather perfect example with the boron transport system, we think the concept they develop might be more widely applicable to a range of other nutrients. The authors should discuss other elements for which such a rapid reporter regulation might be necessary (those where coupled-transcellular transport might occur and where toxicity is an issue)

4) Figures 1 and 2. The labeling is exceedingly poor. There is nothing in Figure 2 to allow the reader to know what C1-C4 etc. are (eventually, we work this out). Please add this to the figure legend. It seems that some of the cells have an identity (C3 is endodermis?) but it is not possible to determine this from the figures. What is the gap between W2 and Wn-2? Please add suitable explanation.

5) Although we think that the reduction of radial transport into a cell file is acceptable in the context of this work, we do not understand why the authors chose a four cell model in which the endodermis is the last cell before the xylem (if we understood it correctly). This is clearly not the case, as the endodermis never touches xylem vessels directly, but is always at least separated from it by on layer of pericycle, which might be quite relevant when considering xylem loading. Please provide some explanation for this.

6) Connected to the above, we think the authors stretch it a bit far when they claim that the observed tissue-specific transporter accumulations can be largely explained by feedback regulation of otherwise constitutively expressed transporters. This might work for *NIP5;1*, but we question whether it can account for the complex accumulation pattern of BOR1 ? We don't think the statement in the text « in the cortex (cell 2) at high levels, whilst in the endodermis it is weaker (cell 3) » reflects what can be seen on the pictures in Figure 1 where BOR1 looks stronger in the endodermis than in the cortex. Please consider providing additional support for Figure 1 (other images?) and discuss these aspects.

7) The experiment in Figure 7 does not evaluate the outcome of the model. Instead, it further evaluates the mechanisms of NIP5 RNA regulation. This is also an interesting question but previous analyses suggested that NIP5 mRNA degradation involved ribosome-stalling controlled by uORFs in the 5' UTR. So how are the current data in Figure 7 reconciled with previously published data in Plant Cell (the Plant Cell data are also generated by some of the authors of this manuscript)?

8) This model has been reported previously (Shimotohno et al., 2015) although it tested different aspects of the system. We are surprised by this omission and request that the authors insert some discussion of the earlier work and its relationship to the current data.

9) The mathematical aspects are very well done with the exception of one small typo in Equation 1B where *W_i-1_* should be *W_i_*probably. Please check this.

10) The authors did a robustness analysis to show that the results are not specific to the parameter set they used, but this should be mentioned more explicitly in the text.

---

## [Author Response]

Essential revisions:1) The authors hypothesize complex relationships between B concentration and NIP or BOR protein abundance and thus activity (Equation 5). However, they may oversimplify the relationship between B concentration and transport itself. Proportionality of transport to B concentration gradients (Equations 1 and 2) is consistent for channels such as NIPs. Is it always applicable to BOR1, assuming that this transporter may saturate at high C_i_?

We here assume a linear relationship between the transmembrane flux due to the activity of the BOR1 transporter and the intracellular boron concentration. We agree with the referee that at a certain concentration this process should saturate and this assumption will not hold anymore. We have experimental reasons, however, to believe that this saturation is not occurring within the concentration range considered here. (The experimental justification is described below, answering the next reviewers’ comment). Nevertheless, we found it both important and interesting to investigate what effects such a saturation in the BOR1-driven transport would have on the system, in particular to its tendency to present traffic-jam like behaviour. We therefore altered the BOR-mediated flux (*J*) as a function of BOR protein levels at the membrane (*B*) and the intracellular boron concentration (*C*), from the standard linear regime (*J* = *BC*) into a Michaelis-Menten saturating form (J=BC1+C/cB, in which *c_B_*is the

Michaelis constant, the boron concentration at which the flux becomes half as large due to saturation, as well as the concentration at which the flux reaches its half-maximum value. We found that as long as *c_B_*is not unrealistically low, the traffic-jam-like behaviour occurs at very similar slowed-down transporter response regimes, but the impact of the traffic jams becomes much more severe, with the concentration fluctuations greatly increasing in amplitude (for *c_B_*= 1000µm, as now shown in the paper, peak values more than double in respect to the non-saturable peak values). We present the increased detrimental effects on the traffic jam when saturation is considered in a supplementary figure. We have however decided not to use such a term for all simulations presented throughout the paper. Firstly because the linear assumption over the relevant regime is not unrealistic (see below); secondly, to avoid introducing an ad-hoc parameter, its value likely to be too low; and thirdly, to prevent the false impression that the traffic-jam-like behaviour is due to or dependent on such a saturation.

The authors should know the likely K_m_ of BOR1 (In any case, the bor1-1 mutant lacks a saturating B uptake component): Under steady state conditions (without oscillation), estimated C_i_ seems to go up to 250 μM and in oscillating conditions, it goes up to 1.5 mM. In the latter conditions, BOR1 must clearly saturate? In our opinion, the authors should explore this important point.

We agree that this is a very interesting point (see above). We believe, however, that a linear relationship is not unreasonable, given what we currently know regarding the saturation in BOR1 transport. Our view is based upon experimental data where expression of *Arabidopsis* BOR1-GFP fusion protein was determined for BY-2 tobacco cultured cells for a range of boric acid concentrations (Yamauchi et al., F1000Research 2013, 2:185). Although the following data was not included in that paper, we measured as part that research the concentration of boron in cells expressing BOR1-GFP fusion. When BY-2 cells expressing BOR1 were exposed to 3, 5 and 10 mM boric acid, intracellular boron concentrations were reduced by ˜10% compared to wild type BY-2 cells, irrespective of the boron concentration in the media. This suggests that the boron transport capacity of BOR1-GFP fusion in BY-2 cell is not saturated in that specific mM concentration range, and thus, we would not expect it to be saturated under the much lower concentrations that our in silico tissues are exposed to.

2) We wondered why the authors hypothesize such a huge difference (30-fold) between k_B_ and k_N_. Is it consistent with actual variations of intracellular B concentration? Data from Figure 1—figure supplement 2 refer to external B concentration and intracellular B may not vary so much. Could the authors present a sensitivity analysis of their model to k_B_/k_N_ ratio?

Firstly, we do believe these values are reasonable given the data. The referees note that “Data from Figure 1—figure supplement 2 refer to external B concentration and intracellular B may not vary so much”. It is reported that the boron concentration in root cell sap is almost linear to that in medium, at least within the range of medium boron concentration from 0 to 100µM (Takano et al., 2010).

For the medium boron concentration higher than 100µM, there are no reported direct measurements of cell sap boron. We therefore instead assume that for the higher external B concentrations presented in Figure 1—figure supplement 2, the concentration in the medium should not to be too far off from the concentrations within and sensed by the outermost cells, in its turn regulating their expression.

Nevertheless, whilst by default using the values for *k_B_*and *k_N_*that are in accordance with Figure 1—figure supplement 2 and the considerations above, we address the concern that the intracellular concentrations in the outermost cells might actually be lower than the concentrations in the medium, giving rise to an overestimation of *k_B_*, by strongly reducing the value of *k_B_*and analysing its impact on the dynamics. The traffic-jamlike behaviour is marginally affected when *k_B_* is reduced from 600 to 400µM, the value we also use for the five-cell-file scenario (see below). When *k_B_*is reduced further the amplitude of the oscillations get smaller, while for *k_B_*= 100µm no oscillations occur, driven by a dramatic (5-fold) reduction in the boron throughput through the tissue. Clearly, traffic-jam-like behaviour can be prevented by a dramatic reduction in default throughput, but this does not seem to be a very adaptive solution. (These results also presented in a figure supplement).

3) While the authors have chosen a rather perfect example with the boron transport system, we think the concept they develop might be more widely applicable to a range of other nutrients. The authors should discuss other elements for which such a rapid reporter regulation might be necessary (those where coupled-transcellular transport might occur and where toxicity is an issue)

We thank the reviewers of pointing this out. Now that we understand the intrinsic instabilities that the boron system is avoiding, we recognise a posteriori that similar temporal constraints are likely to be operating in a wider set of polar substrate-dependent transport systems, such as in phosphate and iron uptake, and even in auxin transport. In the case of phosphate transport, phosphate transporters LePT1 and LePT2 localise to the outer side of the epidermis to facilitate uptake (Liu et al., 1998), and it was recently suggested that phosphate transporters may carry a phosphate sensing domain that can trigger rapid degradation in a phosphate-dependent manner (Gu et al., 2016). Similar regulation has also been reported for the Fe transport system, where polar accumulation of IRT1 (a major Fe transporter for Fe uptake into roots) is regulated in an Fe-dependent manner (Barberon et al., 2014). We now more explicitly comment about this in the manuscript, as well as regarding the possibility that if under certain conditions auxin were to be exerting inhibitory regulation in the transport efficiency of AUX/LAX, PCP and/or PIN family members, it could also, within a polarised tissue, lead to intrinsic transport, and hence auxin, oscillations. Given that oscillations in auxin have received a significant amount of attention during the recent years, this might provide a potential alternative mechanism underlying these patterns.

4) Figures 1 and 2. The labeling is exceedingly poor. There is nothing in Figure 2 to allow the reader to know what C1-C4 etc. are (eventually, we work this out). Please add this to the figure legend. It seems that some of the cells have an identity (C3 is endodermis?) but it is not possible to determine this from the figures. What is the gap between W2 and Wn-2? Please add suitable explanation.

We apologise for this oversight. Our revised manuscript includes clearer figures and figure legends.

Figure 1: We have now added cell identity labels as well as a simple schematic that explains the mapping of each biological cell to those included in our model.

Figure 2: We have now added an explanation for each variable in the figure legend.

5) Although we think that the reduction of radial transport into a cell file is acceptable in the context of this work, we do not understand why the authors chose a four cell model in which the endodermis is the last cell before the xylem (if we understood it correctly). This is clearly not the case, as the endodermis never touches xylem vessels directly, but is always at least separated from it by on layer of pericycle, which might be quite relevant when considering xylem loading. Please provide some explanation for this.

Indeed, as suggested above, our four-cell model is meant to serve as a concise representation of a generic cross-section, and allows for a quicker visualisation of the dynamics of traffic jam behaviour in the concentration-time plots. Even this concise version of the model already contains 14 coupled ODEs, and, as the previous comment pointed out, this can already be overwhelming and requires careful attention to keep the figures readable and understandable. We are therefore reluctant to choose for the main story a more complex version.

Nevertheless, in response to this comment, we have now repeated the main simulations of the paper in a 5-cell context model and added our results as a supplementary figure to further support the robustness of the mechanism. As can be seen in this supplementary figure, 5-cell tissues manifest traffic-jam-like behaviour when transport regulation is lowered in a very comparable way. In fact, as we also now point out, a higher number of cell files actually enlarges the regime for which instabilities are found. Thus, showing the effect in the 4-cell model is not only parsimonious, but also a ‘worst-case’ scenario for these behaviours to manifest themselves.

Moreover, we now emphasise that while the 4-cell model is a “toy” model, and the 5-cell model a “schematic *Arabidopsis*” model, higher number of cell files, as displayed in Figure 3 and the supplementary fig, can represent other plant species.

6) Connected to the above, we think the authors stretch it a bit far when they claim that the observed tissue-specific transporter accumulations can be largely explained by feedback regulation of otherwise constitutively expressed transporters. This might work for NIP5;1, but we question whether it can account for the complex accumulation pattern of BOR1 ? We don't think the statement in the text « in the cortex (cell 2) at high levels, whilst in the endodermis it is weaker (cell 3) » reflects what can be seen on the pictures in Figure 1 where BOR1 looks stronger in the endodermis than in the cortex. Please consider providing additional support for Figure 1 (other images?) and discuss these aspects.

We agree, in contrast to the *NIP5;1* patterning, which is a clear and robust result, the detailed quantification and interpretation of the BOR1 accumulation patterning is not a main focus of this work, but an interesting side observation that we made from the model. We have now altered the text so that this observation will not be over-interpreted and obscure the main findings of the model. These changes have been made to the relevant section, now entitled: “Parsimonious model, under wild type dynamical settings, generates stable transporter expression”.

7) The experiment in Figure 7 does not evaluate the outcome of the model. Instead, it further evaluates the mechanisms of NIP5 RNA regulation. This is also an interesting question but previous analyses suggested that NIP5 mRNA degradation involved ribosome-stalling controlled by uORFs in the 5' UTR. So how are the current data in Figure 7 reconciled with previously published data in Plant Cell (the Plant Cell data are also generated by some of the authors of this manuscript)?

We appreciate you raising this point. We recently demonstrated that *NIP5;1* mRNA degradation occurs through ribosome stalling and that degradation occurs in the cytoplasm (Tanaka et al., 2016). However, we also report that an additional sequence exists upstream of the AUG-stops that also promotes mRNA degradation, but does not trigger ribosome stalling. The current data shown in Figure 7 (in our submitted manuscript), provides evidence of mRNA degradation occurring close to the site of transcription. Considering all evidence together, we propose that mRNA degradation could occur through two coordinated B-dependent processes; one localised in the nucleus, where RNA is turned over at the site of transcription, and in addition cytoplasmic degradation that is triggered by ribosome stalling. An alternative hypothesis could be that *NIP5;1* mRNA is transported into the cytoplasm though the nuclear pores close to the site of transcription and the transported mRNA is rapidly degraded by the mechanism described in Tanaka et al., 2016. We have not further tested either of the hypotheses as we feel that it lies beyond the scope of this paper. But we now make this suggestion in the revised manuscript and explain how it can reconcile our current data with previously published work.

8) This model has been reported previously (Shimotohno et al., 2015) although it tested different aspects of the system. We are surprised by this omission and request that the authors insert some discussion of the earlier work and its relationship to the current data.

We now mention and explain the main insights of that previous model and thank the reviewers for this suggestion, as those findings complement this work greatly. We have added this explanation to the section that introduces the model and modelling choices.

9) The mathematical aspects are very well done with the exception of one small typo in Equation 1B where W_i-1_ should be W_i_ probably. Please check this.

Thank you for spotting this typo!

10) The authors did a robustness analysis to show that the results are not specific to the parameter set they used, but this should be mentioned more explicitly in the text.

We now do so, and also have further increased our understanding of the robustness of this phenomenon through the new supplementary simulations that address the points raised above (concerning transport saturation, *k_B_/k_N_*relationship, and choice of root file number). We now mention explicitly in the text that, taken this all together, there is a structural need of traffic jam avoidance in plants.